

# Estimation of Snow Depth from AMSR-2 Based on an AutoML method over the Qinghai-Tibet Plateau

Xuan Li [1], Fan Xu [2], Chen Zhang [1], Yanli Zhang [1]

[1] College of Geography and Environmental Science, Northwest Normal University, Lanzhou 730070, China

[2] Tencent Dadi Tongtu (Beijing) Technology Co., Ltd., Beijing, China

*Correspondence to*: Yanli Zhang (zyl0322@nwnu.edu.cn)

**Abstract.** Snow depth (SD) is a crucial parameter for describing the spatiotemporal variations of snow cover, and passive microwave SD products (10-25 km) are widely used for monitoring SD changes. However, as one of the three major snow-covered regions in China, the Qinghai-Tibet Plateau (QTP) has complex terrain and rapid change in snow cover with strong

spatial heterogeneity, making it difficult for coarse-resolution SD products to accurately describe its spatiotemporal characteristics. This study proposes a high spatial resolution (500 m) SD estimation method based on AMSR-2 brightness temperature (BT) data and an Automated Machine Learning (AutoML). Firstly, using Pearson correlation coefficient, 19 key factors influencing SD, including AMSR-2 BT, slope, and surface roughness, were selected as input data (independent variables) for AutoML. Meanwhile, a passive microwave downscaled SD data and ground-based SD measurements were

introduced as dependent variables for AutoML. Then, the AutoML model was trained separately for four different types of snow cover surfaces (forest, grassland, water, and bare land). Finally, through the ten folds cross validation method, the optimal machine learning model for SD estimation under each type of underlying surface coverage was selected, thus sequential SD datasets were obtained for ten-year snow cover periods of the QTP from 2012 to 2021. Results show that the estimated SD values are consistent with ground-based observations (R=0.81), and the accuracy is high with an RMSE of

3.65 cm. Compared with Landsat-8, the estimated SD spatial distribution is consistent with the snow cover extent on optical images, which can provide reliable data for monitoring snow cover changes in mountainous regions.

**Keywords:** Snow depth; Automated machine learning; Brightness temperature

## 1 Introduction

Snow cover is one of the key elements of the cryosphere, which is critical for global ecosystems, hydrological cycles, and

human societies. Snow depth (SD) is an important attribute describing the spatiotemporal variation of snow cover, and a crucial parameter in various fields, such as climate change and hydrological cycle. The Qinghai-Tibet Plateau (QTP), known as the "Roof of the World," is one of the three major snow-covered regions in China and a sensitive area for global climate change (Tedesco et al., 2010; Zhang et al., 2008). With global warming, the temperature in the QTP has changed significantly, and snow cover has decreased over the years, with extremely uneven spatial distribution. Especially since the



2000s, the SD on the QTP has shown a significant downward trend, and there are large differences in its spatiotemporal
distribution characteristics (Che et al., 2019; Wang et al., 2022; Yu, 2014). Therefore, monitoring the SD changes on the
QTP is highly significant for meteorological forecasting, water resource management, hydrological modeling, and other
related fields.

Research on SD inversion based on passive microwave remote sensing has been conducted for more than 40 years. Multiple

mature inversion algorithms have been developed, and various SD products have been released. Currently, there are three
main methods for using passive microwave remote sensing to invert SD: physical model method, semi-empirical statistical
method, and machine learning (ML). Among them, the physical model simulates the scattering and absorption characteristics
of snow in microwave bands, fully considering snow properties such as snow density and snow grain size. However, due to
the complexity of the microwave radiation transmission model and the difficulty in accurately obtaining these snow

characteristic parameters, the reliability of SD physical model is reduced.

The SD inversion of semi-empirical statistical method primarily utilizes the correlation between the difference in the snow
scattering characteristics of different frequency brightness temperature (BT) and SD. The "brightness temperature gradient
method", initially proposed by Chang et al. (1987) (Chang et al., 1987), has been widely used. and numerous scholars have
subsequently improved SD inversion algorithms based on Chang algorithm (Cao et al., 1993; Che et al., 2008; Foster et al.,

1997; Jiang et al., 2014; Kelly, 2009). Among them, Che et al. (2008) improved the Chang algorithm based on SD
measurements from Chinese meteorological stations in response to the low snow density in China, and released two long-
term time series SD datasets in China: Che_SSMI/S product and Che_SMSR2 product. In addition, some studies considered
the influence of different snow underlying surface types on the accumulation and spatial distributionthe of snow cover, and
proposed a SD inversion algorithms based on multi-frequency BT data. Jiang et al. (2014) combined four frequencies (10

GHz, 18 GHz, 36 GHz, and 89 GHz) BT data to establish a semi-empirical SD inversion algorithm with four snow
underlayment cover types (grassland, farmland, bare land, and forest).

However, owing to the low spatial resolution (10-25 km) of these passive microwave SD products, the accuracy of SD
inversion is significantly limited in mountainous areas. Some scholars have conducted downscaling research on passive
microwave SD products based on snow cover distribution data. Tang et al. (2016) downscaled the Che_SSMI/S product to

obtain daily SD (0.05°) for the QTP based on 500 m MODIS fractional snow cover (FSC) dataset (2000-2011) using
empirical fusion rules and snowmelt regression curves. Hu et al. (2021) developed a spatially dynamic SD downscaling
algorithm for the northern Xinjiang region using AMSR-2 BT combined with MODIS FSC data, improving the downscaling
accuracy of SD (RMSE=3.47 cm). Xu et al. (2024) conducted a downscaling comparison of two widely used SD datasets
(Che_SSMI/S and Che_AMSR2) using MODIS FSC products. The results showed that the downscaled SD of Che_AMSR2

(Che_AMSR2_NSD) was more consistent with the SD observations.

In recent years, ML has become a significant means of SD inversion. By training ML models, such as Support Vector
Machine (SVR), Random Forest Method (RF), and Artificial Neural Network (ANN), a nonlinear relationship between
microwave radiation BT and SD is established, and the SD inversion accuracy is improved by integrating multisource remote





sensing data (Xiao et al., 2018; Zhong et al., 2021). Yang et al. (2020) proposed a SD inversion algorithm based on RF that

considers multiple factors (BT at different frequencies, geographic location information, and land cover types), but the

accuracy of SD inversion is still limited by the acquisition of prior knowledge of SD. Hu et al. (2021) compared three ML

methods (ANN, SVR, and RF) using five SD products and ground-based SD measurements as prior data, and found. that RF

had the highest accuracy. The aforementioned studies indicated that the SD estimation method based on ML models exhibits

significant advantages, however, they still have two shortcomings in mountainous area. (1) ML methods require a large

number of SD data as training samples, but existing SD products have low spatial resolution, and ground-based

measurements are often scarce and unevenly distributed. (2) Generally, single or several ML models are used to train data

for specific regions, and there are many challenges in data processing, feature selection, and the selection of the best model,

which are accomplished through intuition or trial and error (Du et al., 2020).

To address the issues in ML models mentioned above, the Automated Machine Learning (AutoML) has emerged (Benghzial

et al., 2023). Without human intervention, AutoML can autonomously execute a series of processes, including data

processing and model performance evaluation, and ultimately identify the optimal ML model. The Auto WEKA proposed by

Thorntond et al. (2013) is one of the earliest AutoML frameworks (Kotthoff et al., 2017; Thornton et al., 2012), subsequently

various AutoML frameworks have emerged, such as Auto-Sklearn, TPOT, H2O, and Pycaret (Feurer et al., 2015; LeDell and

Poirier, 2020; Olson and Moore, 2016). Among them, Pycaret provides a simple and easy-to-use interface that not only

selects the optimal model by comparing the performance of multiple ML models, but also combines the prediction results of

multiple models together, improving overall performance and robustness. Therefore, AutoML, specifically the Pycaret

model is expected to be an effective tool for SD inversion, although there are currently few applied studies on this area.

The study proposes a method for estimating SD in in mountainous areas based on the AutoML: Pycaret model. Firstly, the

Che-AMSR2 downscaled SD data and ground-based SD observations are used as input data (dependent variables) for the

Pycaret model, whilst the AMSR-2 BT data and 28 factors, such as slope and surface roughness, are used as independent

variables. Then, a total of 19 key factors were screened using the Pearson correlation coefficient method, and the input

sample data was trained for four snow underlying surface types (forest, grassland, water, and unused land). Finally, the

optimal AutoML model is obtained for each snow subsurface coverage type is subsequently selected to estimate SD on the

QTP. The study employs snow cover products to identify the presence or absence of snow in 500 m pixels. For snow-free

pixels, the SD value is set to 0, while for snow-covered pixels, the proposed SD estimation method is utilized to obtain the

SD values anew.

## 2 Study area and data

### 2.1 Study area

The QTP is situated in the southwestern region of China, renowned as the "Roof of the World" and the "Water Tower of

Asia"(Pu and Xu, 2009). As illustrated in Figure 1, the terrain of the QTP is complex and fragmented, and characterized by



high northwest and low southeast, resulting in extremely spatial heterogeneity of snow cover (Huang et al., 2019; Ke and Li, 1998; Li et al., 2022b). Moreover, the seasonal variation of snow cover on the QTP is pronounced, with the widest distribution in winter, gradually decreasing snow cover in spring and autumn, and the smallest snow coverage in summer. Usually, the snow cover period spans from October to March of the following year, with October to December being the

accumulation period, January to February being the stable period, and March being the melting period (Lu et al., 2008; Qin, 2012).

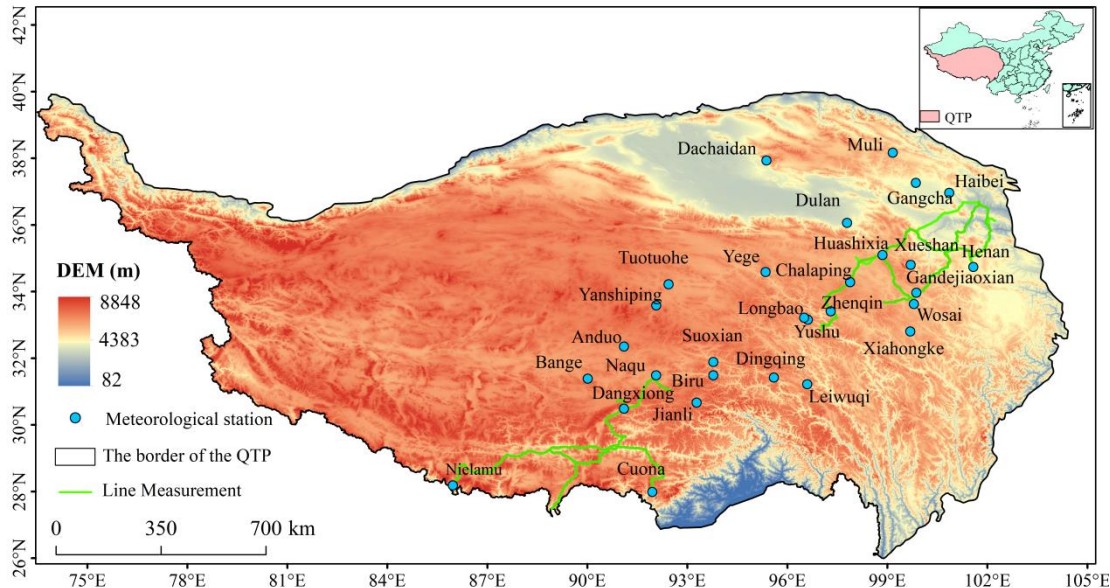

**Figure 1: Location of study area and ground-based SD observations.**

## 2.2 Data Sources and Preprocessing

As shown in Table 1, the dataset used for this research experiment comprises five main categories: AMSR-2 (Advanced Microwave Scanning Radiometer 2) BT; downscaled SD data (Che_AMSR2_NSD); daily cloud-free snow cover products; ground-based SD observations; and other auxiliary data.

**Table 1: Basic Information of the Experimental Dataset.**

| Datasets | | Spatial Resolution | Data period | Data sources | Application |
|---|---|---|---|---|---|
| AMSR-2 BT | | 10 km | 2012.10~2021.03 | https://gportal.jaxa.jp/ | Establish model |
| Che_AMSR2_NSD | | 500 m | 2012.10~2018.03 | - | Input data |
| Daily cloud-free snow cover dataset | | 500 m | 2012.10~2021.03 | https://poles.tpdc.ac.cn/zh-hans/ | Snow Identification |
| SD observations | Meteorological station | - | 2015~2019 | https://data.tpdc.ac.cn/home/ | Input data and verification |



| | Measurement routes | - | 2018～2019 | https://www.csdata.org | |
|---|---|---|---|---|---|
| | | | | https://www.ncdc.ac.cn/ | |
| Auxiliary Data | SRTM DEM | 90 m | - | https://earthexplorer.usgs.gov / | Aspect、slope et al |
| | CNLUCC | 1 km | 2020 | https://www.resdc.cn/ | Land cover types |
| | ERA5-Land | 1 km | 2012.10～2021.03 | https://climate.copernicus.eu/ | Average monthly temperature |
| | Landsat-8 | 30 m | 2012.10～2018.03 | https://www.usgs.gov | Assessment of snow cover |

### 2.2.1 AMSR-2 BT

AMSR-2, a microwave sensor mounted on the GCOM-W1 satellite launched by the Japan Aerospace Exploration Agency (JAXA), conducts observations at seven frequencies, each with horizontal and vertical polarization modes (Imaoka et al., 2012). It has a spatial resolution of 10 km and revisits the QTP region every two days. Some studies have demonstrated that the quality of descending BT data is significantly superior to that of ascending BT data (Huang et al., 2022). Therefore, for this study, descending AMSR-2 L1B data were downloaded for five frequencies (10.65 GHz, 18.7 GHz, 23.8 GHz, 36.5

GHz and 89.0 GHz), including two polarization modes, during the snow cover periods of the QTP from 2012 to 2021. The AMSR-2 data were then resampled to 500 m using nearest neighbour interpolation to extract the BT values corresponding to the SD sample points.

### 2.2.2 Che_AMSR2_NSD

Che_AMSR2_NSD is a 500-m downscaled Che_AMSR2 dataset, which was obtained from the results of a published study

that utilised empirical fusion rules and snowmelt regression curves (Xu et al., 2024). In comparison to the SD data from meteorological stations, it exhibits a higher degree of concordance with measured SD, with an R of 0.72 and a root mean square error (RMSE) of 3.21 cm. Therefore, the Che_AMSR2_NSD, in conjunction with ground-based SD observations, was utilised as a training sample for the AutoML.

### 2.2.3 The daily cloud-free snow cover dataset

The daily cloud-free snow cover dataset is freely available on the Big Earth Data Platform for Three Poles, with a spatial resolution of 500 m and a temporal resolution of 1 day (Huang et al., 2018). The present study sought to ascertain whether there is a distribution of snow in pixels by downloading daily snow cover data from 2012 to 2021 over the QTP during the snow cover periods.

### 2.2.4 SD observations

The ground-based SD observations utilised in this study can be categorised into two distinct types: measurement routes and meteorological stations. The initial step in this research involved the procurement of a comprehensive set of data pertaining



to snow surveys, which was obtained from the "Survey of Snow Characteristics and Distribution in China" project. This data set was derived from measurement routes, providing a detailed and precise overview of the study's subject. The study encompasses three snow survey routes in China from 2018 to 2019, encompassing over 200 manually observed snow sample points (Li et al., 2022a). Secondly, meteorological station observations in SD were obtained from an automatic measurement dataset on the SD in the QTP (2015-2016) (Jiang et al., 2017) and regular stations in typical regions of China during 2017-2019 (Li et al., 2021). The data presented herein were obtained from the China Scientific Data and the National Cryosphere Desert Data Center. The present study utilised all ground-based SD observations on even dates as the input data for the AutoML model, with observations on odd dates serving as the validation data for SD estimation.In this study, all ground-based SD observations from even-numbered dates were selected as the input data for the AutoML model, while data from odd-numbered dates was employed to validate the SD estimation results.

### 2.2.5 Auxiliary Data

The auxiliary data utilised in this study is predominantly categorised into four distinct groups: SRTM DEM, China Multi-period Land Use Remote Sensing Monitoring Dataset, ERA5 Land Temperature Data, and Landsat-8 optical images. The SRTM Digital Elevation Model (DEM) data was generated by the National Aeronautics and Space Administration (NASA) during Earth observation missions. The data has a spatial resolution of 30 metres and is stored in Geo-TIFF format. It is freely available from the United States Geological Survey (USGS). Preprocessing steps such as cropping and resampling were applied to obtain a 500 m resolution DEM dataset for the QTP. The China's Multi-temporal Land Use Remote Sensing Monitoring Dataset (CNLUCC) is derived from Landsat remote sensing images and manually interpreted to produce a dataset with a spatial resolution of 30 m. This dataset is available for download at no cost from the Resource and Environmental Science Data Center. (Xu et al., 2018). The present study utilised the classification results of the aforementioned dataset to calculate the proportion of each land cover type in the QTP, thereby identifying major land cover types such as forests, grasslands, water, and unused land. The establishment of distinct ML models was undertaken for the purpose of estimating the SD of each of the designated land cover types. Landsat-8 optical remote sensing images are utilised predominantly for comparative analysis of SD spatial distribution in the Auto_NSD dataset. These images were obtained from the official website of the United States Geological Survey, with a spatial resolution of 30 m and a revisit period of 16 days. For the purposes of this study, cloud-free images from seven consecutive days were selected as validation data for the Auto_NSD dataset. The ERA5-Land reanalysis dataset is a set of meteorological data concerning the monthly average air temperature at 2 metres above ground level. The monthly average temperature data during the snow season from October 2012 to March 2021 were obtained free of charge from the Copernicus Climate Change Service data platform. These data were then utilised to analyse the SD results obtained from AutoML estimation.




## 3 Methodology

As demonstrated in Figure 2, the estimation of SD based on AMSR-2 BT data and the Pycaret model involves three distinct

steps. Initially, 19 key factors influencing SD, including AMSR-2 BT data, slope, and surface roughness, were selected using

the Pearson correlation coefficient method and designated as input parameters for AutoML. Furthermore, the ground-based

SD measurements and the 500 m downscaled SD data from a passive microwave SD product were introduced as dependent

variables for the model. Subsequently, the AutoML model was trained using the aforementioned input data for four distinct

snow underlayment cover types: forest, grassland, water, and unused land. Ultimately, the optimal ML model for each snow

underlayment cover type was selected through ten-fold cross-validation. Moreover, the spatiotemporal variation

characteristics of SD during the snow cover period on the QTP were obtained from 2012 to 2021. The present study utilised

AMSR-2 BT data and selected influencing factors of SD, such as slope and surface roughness, evaluated through Pearson

correlation coefficients, as independent variables. The Che_AMSR2 downscaled SD and ground SD observation data were

utilised as input data (dependent variables) for the AutoML models. The samples were trained under four different types of

snow-covered surfaces. Finally, ten-fold cross-validation was conducted to assess the performance. The selection of the

optimal ML models was conducted for each category of snow-covered surface, and the SD during the snow cover period on

the QTP from 2012 to 2021 was estimated. The technical roadmap of this study is illustrated in Figure 2.

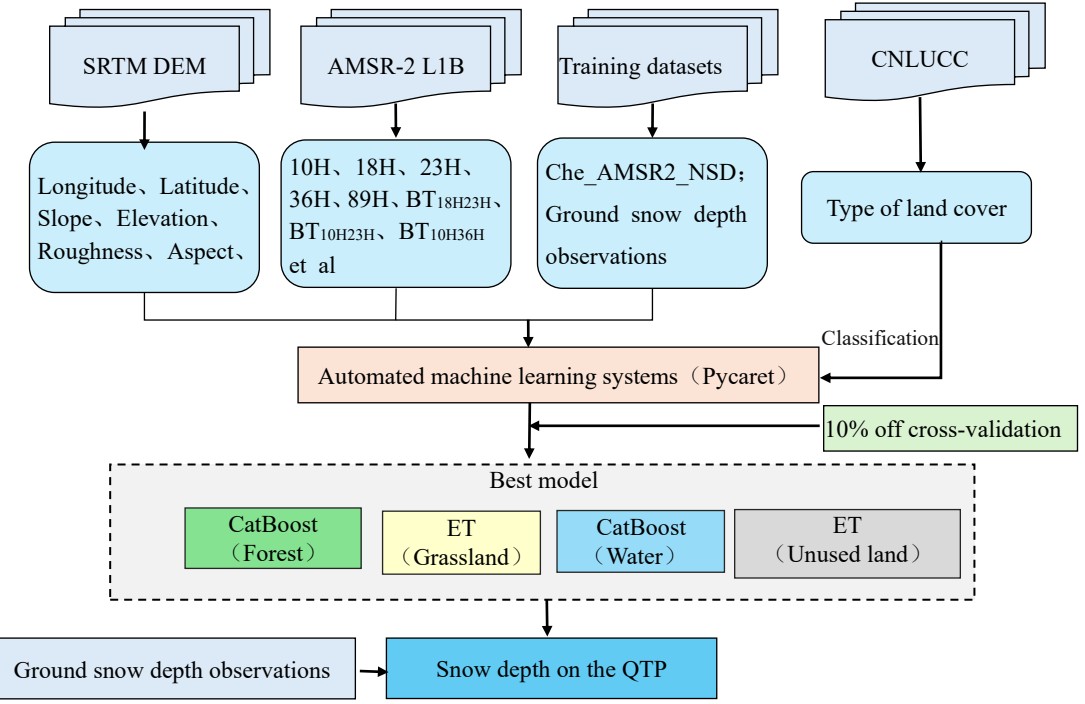

**Figure 2: Flowchart of this algorithm.**





## 3.1 AMSR-2 SD estimation based on AutoML

The present study elected to utilise the Pycaret AutoML framework to execute a series of steps, including data processing and SD model selection. Its workflow involves the generation of multiple models following the optimisation of the hyperparameters of each model based on user-defined inputs and outputs, as well as specific performance metrics (Xu, 2023). It mainly consists of three parts: meta-learning, Bayesian optimization, and model integration (Figure 3). The model is composed of three constituent elements: meta-learning, Bayesian optimisation, and model integration (Figure 3). During the

meta-learning phase, Pycaret continuously refines the learning strategy and model selection by analysing the learning of historical data and the performance of the models to enhance overall performance. Through comprehensive exploration of data features, model algorithms, and hyperparameters, metalearning enables Pycaret to comprehend the complexity of the data and provide more accurate prediction results. Bayesian optimization represents a pivotal technique employed by the Pycaret framework to calibrate model hyperparameters. By leveraging Bayesian optimization, Pycaret intelligently selects

subsequent combinations of hyperparameters to evaluate based on the results of previous model performance assessments. This process efficiently searches the parameter space and accelerates the model optimization process (Silva et al., 2025). In essence, the process of model integration involves the amalgamation of multiple high-performing models into a unified entity. This integration serves to augment the accuracy and stability of predictions.

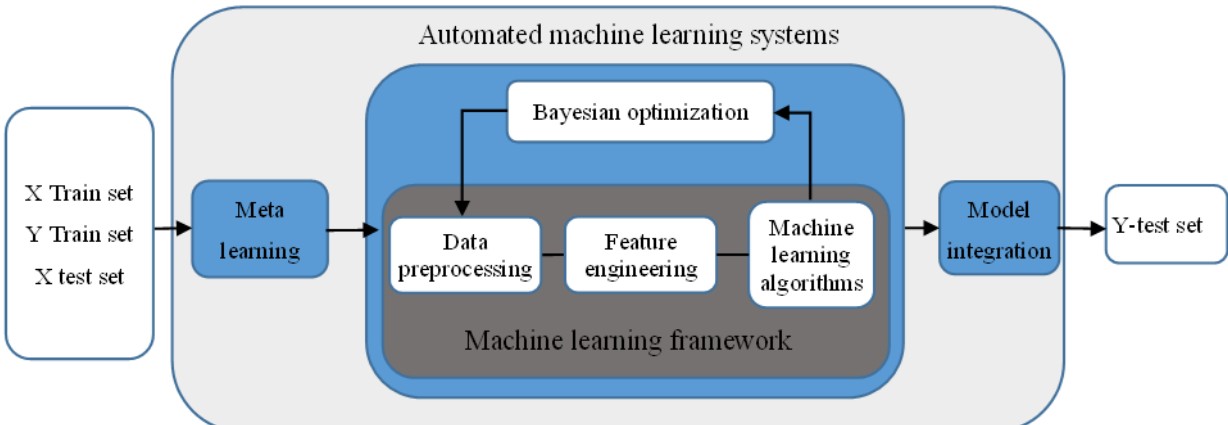

**Figure 3: Flowchart of this algorithm.**

The objective of the Pycaret framework is to reduce the barriers to entry for ML, thereby facilitating a more streamlined and efficient process. This will enable users to compare, select and deploy models with greater ease. Pycaret comprises three primary categories of ML models: namely, generalized linear, tree-based, and ensemble learning models. Linear models encompass a variety of algorithms, including Ridge regression, Lasso regression, Bayesian ridge, Lasso least angle

regression, and Huber regressor. Tree-based models consist of the following elements: the Decision Tree Regressor, the Random Forest Regressor, and the Extra Trees Regressor. It is evident that ensemble learning models are predominantly





composed of gradient boosting regressors, XGBoost, light gradient boosting machines, and CatBoost regressors (Silva et al., 2025; Xu, 2023).

### 3.1.1 Key factor selection

The SD inversion is affected by multiple factors, and initial research focused on the sensitivity of various microwave frequencies to snow cover. The SD inversion was carried out by using the BT values of each microwave frequency. Chang's algorithm is chiefly reliant on BT data from 18 GHz and 36 GHz in order to derive SD. Nevertheless, in regions characterised by shallow snow cover, the SD inversion results obtained using this algorithm demonstrate poor performance (Chang et al., 1987). Consequently, a number of scholars have employed the BT data supplied at 89 GHz, 23 GHz and 10
GHz in the context of SD inversion studies (Jiang et al., 2014; Kelly, 2009; Yang et al., 2020a). Nonetheless, recent research has demonstrated that, in addition to the BT values at different frequencies, geographical location and topographic conditions also exert a significant influence on SD inversion (Huang et al., 2019; Wei et al., 2021). In order to consider the aforementioned factors in a more comprehensive manner, the present study is based on the BT(10H/V, 18H/V, 23H/V, 36H/V, 89H/V) and BT difference data (18H23H, 18V23H, 10H36H, 10H23H, 10V23H, 10V23V, 23V23H, 10V36H,
36H89H, 36V89V, 18H36H, 18V36V). The following additional geographical parameters are to be considered: longitude (Lat), latitude (Lon), elevation, slope, aspect and surface roughness (roughness). The study incorporated a comprehensive set of 28 SD influencing factors. In order to evaluate the interrelationship between the respective variables in depth, the Pearson correlation coefficient (r) was used to analyse them. The coefficient was calculated using the following Eq. (1):

$$r = \frac{\sum_{i=1}^{n}\left(Xi - \overline{X}\right)\left(Yi - \overline{Y}\right)}{\sqrt{\sum_{i=1}^{n}\left(Xi - \overline{X}\right)^2}\sqrt{\sum_{i=1}^{n}\left(Yi - \overline{Y}\right)^2}} \tag{1}$$

Where r is the Pearson correlation coefficient, $X_i$ and $Y_i$ represent the sample of the two independent variables, and $\overline{X}$ and $\overline{Y}$, respectively, represent the average value of each independent variable, while n represents the number of samples. The value of r ranges from -1 to 1. The presence of a strong positive or negative correlation between two variables is indicated by a value of R greater than 0.90 or less than -0.90, respectively.

### 3.1.2 Model selection and construction

The distribution of meteorological stations across the QTP is characterised by a relative paucity of stations, with manual field SD data being limited in scope and predominantly concentrated in the eastern region. Moreover, the preponderance of meteorological stations is located on grassland or unutilised land surface types. Consequently, the utilisation of solely ground SD observation data as sample data for AutoML may not ensure sufficient representativeness. In order to solve this problem, in addition to the observation data of ground SD, 471 sample points (Figure 4) were selected from the Che_AMSR2_NSD



data. In selecting the sample points, the influences of slope direction, elevation and gradient of different terrain conditions were comprehensively considered, and ensured to be uniformly distributed across the QTP. Furthermore, the subsurface types have been demonstrated to exert a substantial influence on the aggregation and distribution of snow cover, which in turn has a considerable impact on the accuracy of SD inversion. The resolution of AMSR-2 passive microwave remote sensing pixels is coarser (10 km), which gives rise to a more significant problem of mixed pixels. This problem constitutes

one of the primary sources of error in SD inversion (Jiang et al., 2014). Distinctive algorithms for SD inversion have been developed by a number of foreign scholars for a range of land cover types (Derksen et al., 2005; Goïta et al., 2003; Jiang et al., 2014). For example, Derksen et al. (2005) established an inversion algorithm for the predominant land cover types when inverting SD in the forested regions of Canada (Derksen et al., 2005). They subsequently calculated the SD under mixed image elements. Meanwhile, Jiang et al. (2014) established a semi-empirical statistical inversion algorithm for SD in China

under different land cover types. The present study investigates the impact of mixed pixels on the precision of SD inversion, and establishes an AutoML model under various land cover types.

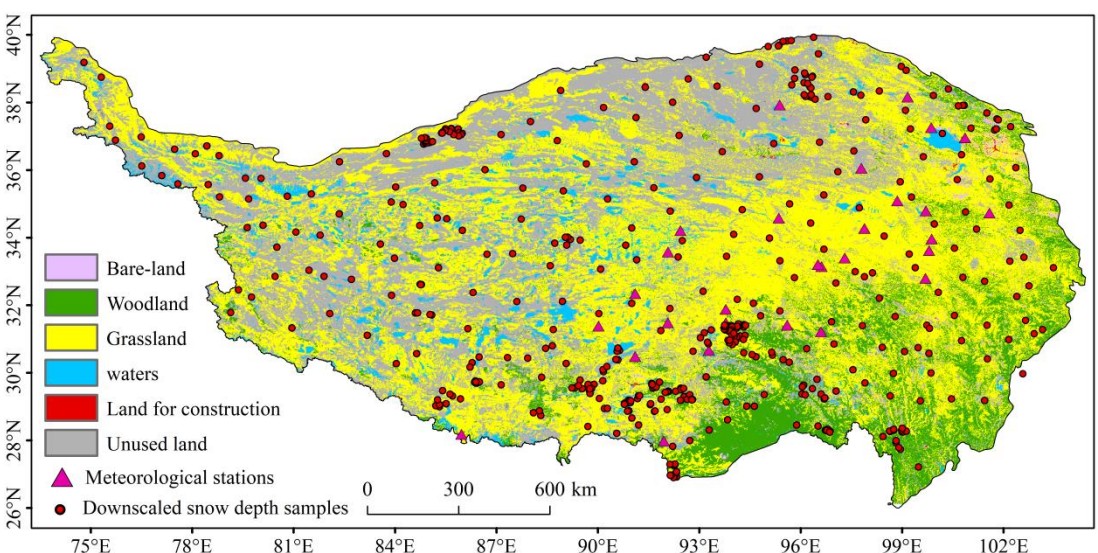

**Figure 4: Spatial distribution of input sample data from the AutoML.**

In this study, 60 forests, 80 water bodies, 171 grasslands and 160 unused land were selected with reference to the major land

cover types on the QTP. Initially, the SD data corresponding to each sample point and 19 influencing factors, such as BT, bright temperature difference and topographic characteristics evaluated and screened by Pearson correlation coefficient, were extracted. In the event that any of the factors were missing, all the corresponding data were eliminated. The final selection comprised 25,926 forest land samples, 340,326 grassland samples, 157,252 water samples and 273,672 unused land samples. The entire sample was subjected to an AutoML system, and the random search function was used to identify the optimal

parameters of various algorithms. These were subsequently employed to assess the accuracy of each ML model under each land cover type using ten-fold cross-validation. The machine is programmed to automatically partition the training set and



the test set, with the training set comprising 90% of the total number of samples and the test set comprising 10% of the total number of samples. The average value of the accuracy evaluation index is then taken to describe the performance of the model after 10 tests. Finally, the most ML model was employed to simulate the snowpack period of the QTP from 2012 to 255    2021.

## 3.2 Accuracy evaluation method

In this study, the performance of the ML model was evaluated by using ten-fold cross-validation, which entails the random division of the original dataset into 10 equal-sized subsets. Thereafter, one subset is selected as the test dataset, with the remaining nine subsets being designated as the training dataset for each cross-validation. The model is then trained on the 260    training dataset, and the model performance is evaluated on the test dataset. This process is repeated ten times, following the previously outlined steps. For each iteration, a distinct test dataset should be selected, with each subset of the data serving as a test set. In conclusion, a comprehensive evaluation of the results of each test is conducted, with the mean value typically serving as the performance index of the model. This evaluation aims to ascertain the model's accuracy and reliability. Three evaluation indexes, the $R^2$, the RMSE and the mean absolute error (MAE), were selected to evaluate the performance of each 265    ML model.

Four precision evaluation indexes, namely $R^2$, RMSE, BIAS and MAE, were selected for the purpose of quantitatively analysing the SD results estimated based on AutoML. R2 and R are utilised to evaluate the regression model's capacity to explain the variations in the dependent variable. These metrics range from 0 to 1, with higher values indicating a stronger correspondence between the model and the data. RMSE is defined as the standard deviation of fit in the regression system, 270    which quantifies the average distance between the predicted value of the model and the actual value; MAE is the mean absolute difference between the predicted and actual values of the model; and BIAS is the positive or negative bias of the SD inversion results, where a smaller absolute value indicates higher accuracy in SD estimation. RMSE, MAE, MAPE and BIAS are metrics that quantify the discrepancy between observed and predicted values. It is generally accepted that lower values for these metrics are indicative of superior model performance.

## 275    4 Results

### 4.1 Evaluation of SD estimation model

#### 4.1.1 Factor selection results

The study utilised a Pearson correlation coefficient analysis on 28 independent variables, the results of which are presented in the form of a heatmap (Figure 5). This facilitates intuitive visualisation of the relationships between variables based on 280    colour intensity. The figure demonstrates a strong correlation between the BT of horizontally and vertically polarized values at identical frequency bands, with correlation coefficients exceeding 0.9. Therefore, in order to mitigate the impact of




autocorrelation between variables on the SD estimation model, one of the BT data at the same frequency but with different polarizations was removed. Additionally, robust correlations were observed between 10H23H & 10V23H, 36H89H & 36V89V, 10H36H & 18H36H, and 10H36H & 18V36V, all exceeding 0.95. In order to ensure the accuracy of the model,

285 variables with correlation coefficients greater than 0.90 were removed. The voltages under consideration are 10V, 18V, 23V, 36V, 89V, 10H23H, 10H36H, 18H36H and 36H89H. In conclusion, a total of 19 independent variables were selected for utilisation as input data for the AutoML model. The dependent variable, SD data, was applied in conjunction with the model during the training process.

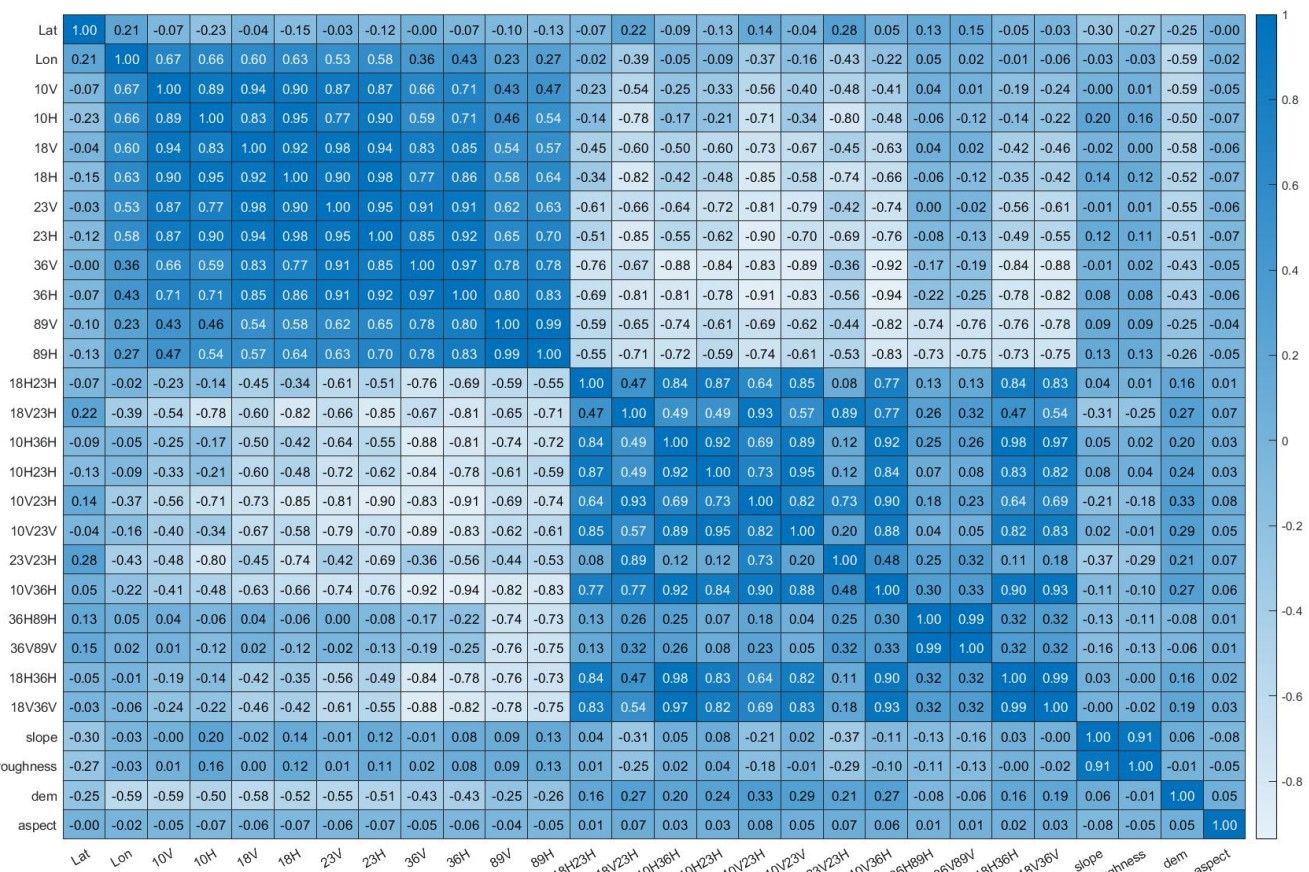

290 **Figure 5: Heat map of Pearson correlation coefficient analysis results.**

### 4.1.2 Model selection results

The study incorporated a total of 25,926 forest samples, 340,326 grassland samples, 157,252 water samples, and 273,672 unused land samples into the AutoML models. The study evaluated the accuracy of each ML model for each land cover type using ten-fold cross-validation. The results of three accuracy evaluation metrics ($R^2$, RMSE, and MAE) for ten ML models

295 (ET, RF, XGBoost, Catboost, LightGBM, KNN, GBDT, LR, Lasso, DT) across four land cover types are presented in Figure





6 (a), (b), and (c). It is evident that in forest regions, the XGBoost model demonstrates the highest level of accuracy ($R^2$ = 0.71, RMSE = 3.30 cm, MAE = 2.24, MAPE = 0.52), followed by the Catboost and LightGBM ensemble learning models, both of which achieve $R^2$ values in excess of 0.7. In grassland regions, the Catboost model demonstrates the highest level of accuracy, with an $R^2$ of 0.77 and RMSE of 3.11 cm, followed by the RF model, which attains an $R^2$ of 0.76. The XGBoost

and ET models also demonstrate relatively high accuracy, with $R^2$ values exceeding 0.75. In water regions, the ET model demonstrates the highest level of accuracy, with an $R^2$ of 0.75 and RMSE of 2.20 cm, while four other ML models achieve $R^2$ values in excess of 0.70: RF, Catboost, XGBoost, and LightGBM. In regions where land use is minimal, the Catboost model demonstrates the highest level of accuracy, with an $R^2$ of 0.82, closely followed by the ET model. The XGBoost, RF, and LightGBM models also demonstrate good simulation accuracy and minimal errors, with $R^2$ values all exceeding 0.80. In

general, linear models demonstrate comparatively reduced accuracy in comparison to gradient boosting models and tree-based models.

In conclusion, the SD estimation accuracy of each ML model exhibits variation across different land cover types, with the best model training results observed in unutilized land and the worst results in forested regions. Concurrently, the simulation accuracies of the integrated learning models (Catboost, XGBoost) and the tree models (ET, RF) demonstrate superior

performance, whereas the linear model exhibits comparatively lower accuracy. Consequently, extreme gradient boosting tree models are employed to estimate SD in forest and water regions, while CatBoost integrated learning models are utilised for SD estimation in grassland and unused land regions over the QTP.



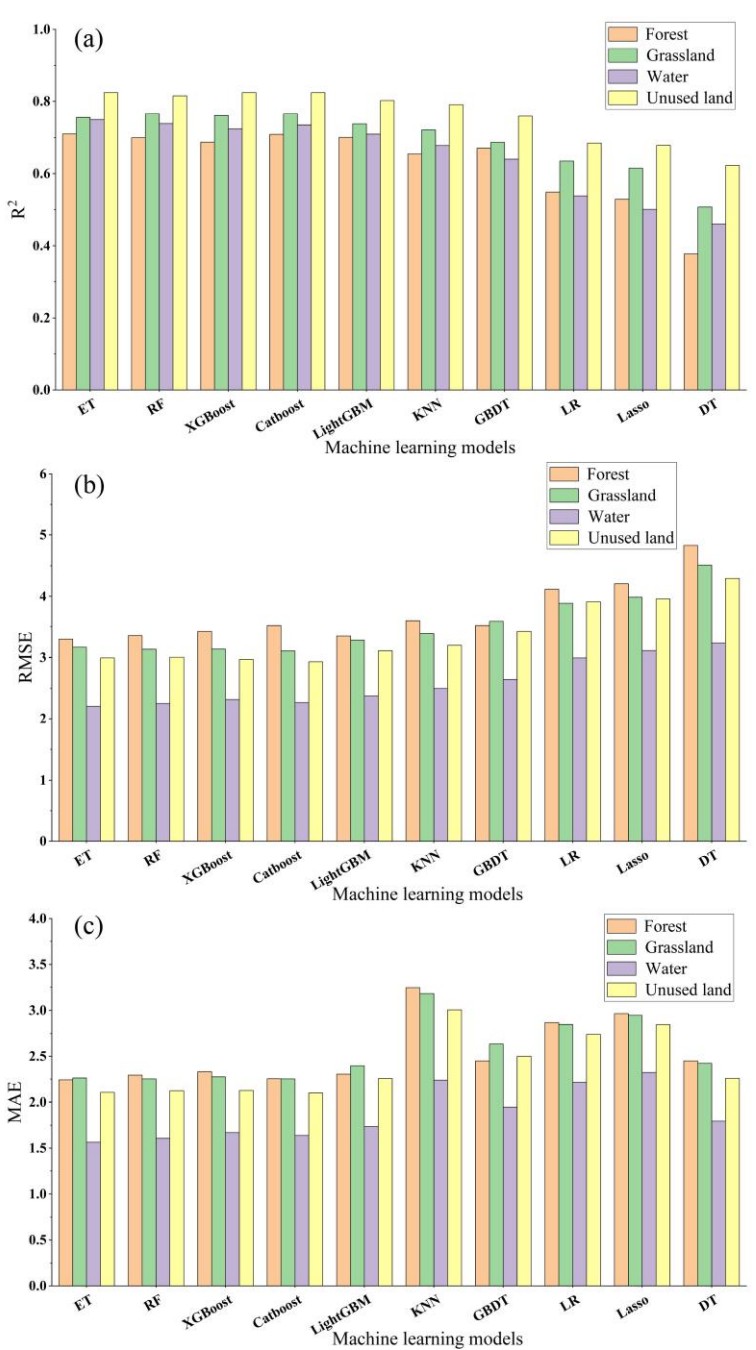

**Figure 6: The results of the accuracy evaluation index of each model under four land cover types: (a) $R^2$, (b) RMSE; (c)MAE.**

### 4.2 Evaluation of the accuracy of SD estimation

This study integrated AMSR-2 passive microwave BT data and geographical factors (longitude, latitude), terrain conditions (slope, aspect, elevation, surface roughness), and other SD influencing factors to develop AutoML models for SD estimation



in forest, grassland, water, and unused land regions. The study was based on the Che_AMSR2_NSD dataset. Subsequently, the optimal models for each land cover type were utilised to estimate 500m SD data (Auto_NSD) over the QTP for nine
snow seasons from October 2012 to March 2021, encompassing 1,603 days. The present study utilised ground-based SD observations for the purpose of conducting a quantitative analysis of the accuracy of Auto_NSD results. Furthermore, the spatial distribution of snow cover was analysed on the basis of extent by means of a qualitative analysis of Landsat-8 optical imagery captured under clear-sky conditions.

### 4.2.1 Evaluation of the overall accuracy of SD results

In order to evaluate the accuracy of the SD results estimated using AutoML, 432 SD data from odd dates of the QTP Meteorological Station SD dataset were utilised as verification data. Four quantitative metrics, namely R, RMSE, BIAS, and MAE, were selected for the accuracy analysis of the SD data obtained from 2012 to 2021. The results of the accuracy validation process are illustrated in Figure 7, demonstrating a high level of consistency with meteorological station SD measurements, with an R value of 0.84. The root RMSE is 3.64 cm, indicating a slight underestimation (bias = -1.72).
Nevertheless, the MAE is comparatively negligible (MAE = 2.93), signifying a high degree of accuracy in SD estimation based on AutoML.

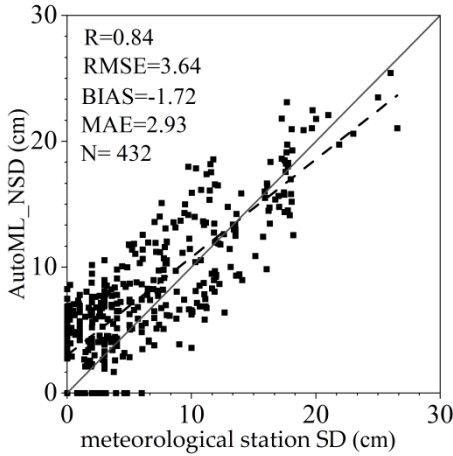

**Figure 7: Scattered plot of SD observed by meteorological stations and SD estimation based on AutoML.**

The present study is based on a survey project that focused on the characteristics and distribution of snow cover in China.
The study obtained a dataset on snow cover observations from typical snow cover regions in China. The QTP primarily comprises survey data from three routes from 2018 to 2019, and the SD measured by 49 routes is compared with the SD estimation based on AutoML as the "true value". As demonstrated in Figure 8, the statistical outcomes exhibit minimal discrepancies and consistent trends between the Auto_NSD data and SD measurements along the surveyed routes. The mean SD for Auto_NSD is 12.77 cm, with a maximum depth of 38.88 cm, whereas the mean depth from route measurements is
14.55 cm, with a maximum depth of 33 cm. Of the sample points examined, 30 exhibited underestimations, accounting for





61% of the cases. The maximum underestimation error was observed on 20 January 2019, at 21.93 cm, while the maximum overestimation error was 8.22 cm on 18 January 2019. The smallest error recorded was 0.15 cm, on 7 December 2018.

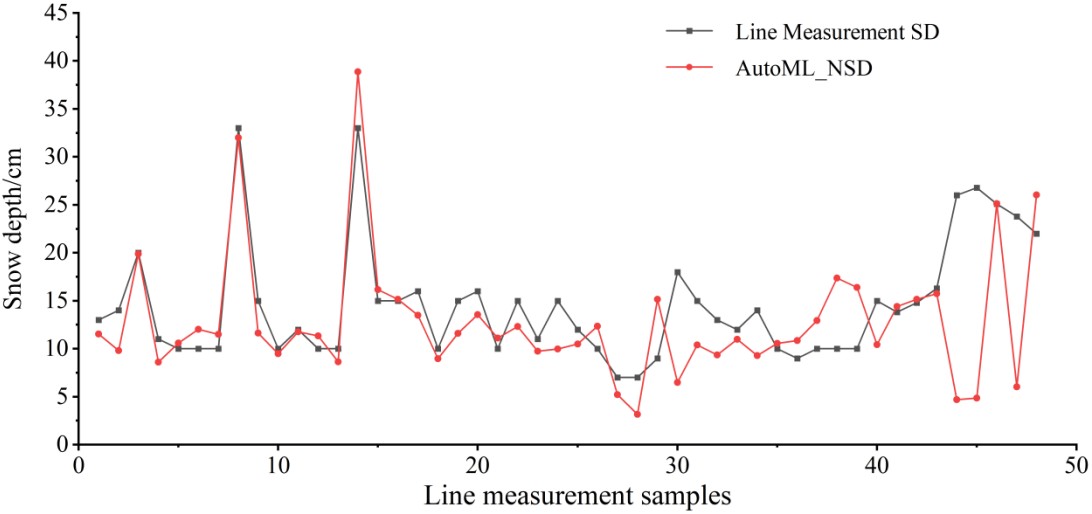

**Figure 8: Line chart of SD estimation and line measurement SD based on AutoML.**

**4.2.2 Evaluation of the spatial accuracy of SD results**

In this study, meteorological station SD observation data was utilised as a reference to calculate the mean SD at each station and the corresponding mean SD from Auto_NSD data. The accuracy of the Auto_NSD data was subsequently analysed using two precision evaluation metrics, RMSE and BIAS. As demonstrated in Figure 9, subplot (a) and (b) illustrate the mean SD for meteorological stations and Auto_NSD data, respectively. Among the meteorological stations where the
discrepancy in mean SD is within 5 cm, 63% were categorised within this range. It is noteworthy that stations such as Henan, Yege, and Zhenqin demonstrate average SD disparities of -0.23 cm, -0.34 cm, and 0.43 cm, respectively, when juxtaposed with Auto_NSD data. The most substantial average standard deviation difference is observed at Niela station, reaching 23 cm. In March 2019, the observed SD at Niela station was notably higher (up to 120 cm) than the approximately 30 cm simulated by the AutoML model, a discrepancy that may be attributable to the original input data of the ML model. The
maximum SD values recorded at sample points in close proximity to the Niela station did not exceed 50 cm. With the exception of this particular station, the mean SD difference at other stations does not exceed 8 cm. Subplot (c) and (d) illustrate the RMSE and BIAS, respectively, between the Auto_NSD data and the meteorological station SD. Stations including Henan, Yegor, Muli, Yanshiping, Haibei, Biru and Dulan demonstrate superior accuracy in SD results, with RMSE values less than 5 centimetres. With the exception of Niela station, the Jiali and Cuona stations demonstrate the lowest levels
of accuracy in SD measurements, with RMSE values of 10.92 cm and 13.95 cm, respectively. A spatial analysis reveals a marked underestimation at Niela station, with Cuona station exhibiting a BIAS of -7.23. Stations such as Zhenqin, Wosai, Henan, and Yegor present SD values that are more closely aligned with those of Auto_NSD data, with BIAS values ranging





from -0.5 to 0.5. A significant correlation between spatial accuracy and SD values is observed through spatial accuracy analysis of SD on the QTP.

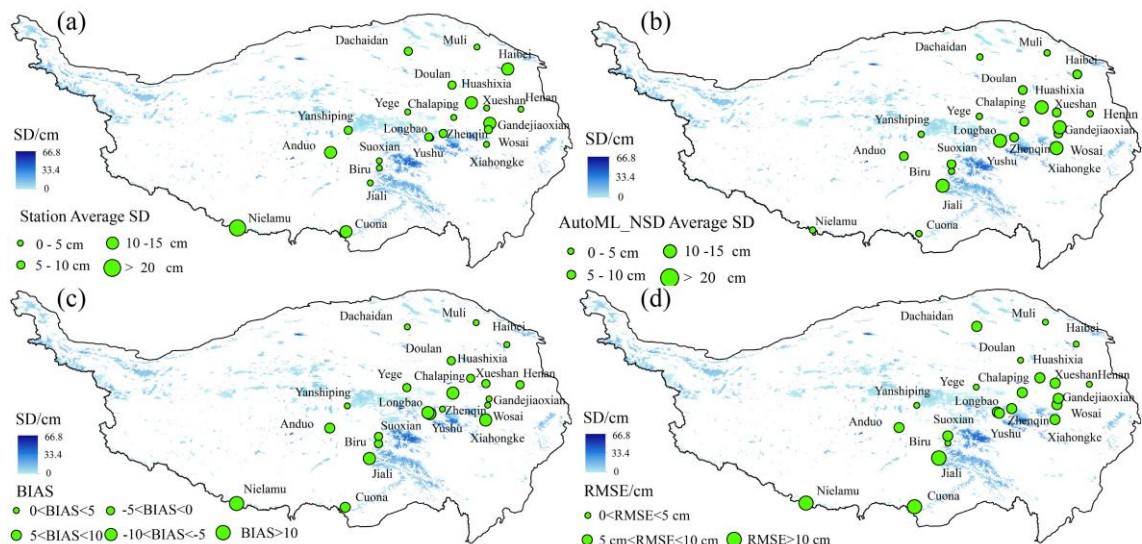


**Figure 9: Spatial error distribution between Auto_NSD data and observed SD at meteorological stations: (a) average SD at meteorological stations; (b) the average SD of Auto_NSD data; (c) Auto_NSD data and RMSE of SD at meteorological stations; (d) Auto_NSD data and the BIAS of SD at meteorological stations.**

In order to analyse the spatial distribution of snow cover in the QTP region at different times and in different representative

terrain regions in more detail, this study selected six cloud-free Landsat-8 satellite images based on their transit times. These images were selected based on the SD results estimated by AutoML. Subsequently, a comparison was conducted between these images and the corresponding Auto_NSD data, alongside downscaled SD data. The results are illustrated in Figure 10, where the colour cyan is used to denote snow cover and red is employed to represent non-snow regions in Landsat-8 images. The figure indicates significant variations in snow distribution across different temporal periods. Specifically, the analysis of

Landsat-8 images from 5 March 2013, 16 November 2015 and 9 November 2016 reveals relatively low snow cover, predominantly concentrated in lower-altitude regions. In contrast, analysis of Landsat-8 images from 5 January 2014, 19 November 2017 and 25 February 2018 revealed higher snow cover, predominantly in high-altitude mountainous regions. In addition, it is evident that both the SD results and the snow distribution observed in the Landsat-8 images are closely correspondent. Nevertheless, discrepancies have been observed between Auto_NSD data and downscaled data in specific

regions. For instance, on 16 November 2015, the Auto_NSD data reflected the actual snow distribution more accurately, while the downscaled SD data exhibited rasterization issues. Furthermore, it was observed that utilising downscaled SD data on 5 January 2014 and 19 November 2017 resulted in the manifestation of more pronounced rasterisation issues in comparison to the utilisation of Auto_NSD data. This discrepancy may be attributed to the comprehensive consideration of multiple SD influencing factors in the AutoML model, resulting in more precise adjustments of pixel SD values. In summary,

both SD estimated by AutoML and downscaled SD data can depict detailed characteristics of SD spatial distribution in




mountainous regions. However, the former provides a more consistent representation of snow distribution in accordance with observed conditions, which may be attributable to limitations in the accuracy of the original passive microwave SD product influencing the precision of downscaled SD results, although it substantially mitigates the issue of blackness in the original SD product.

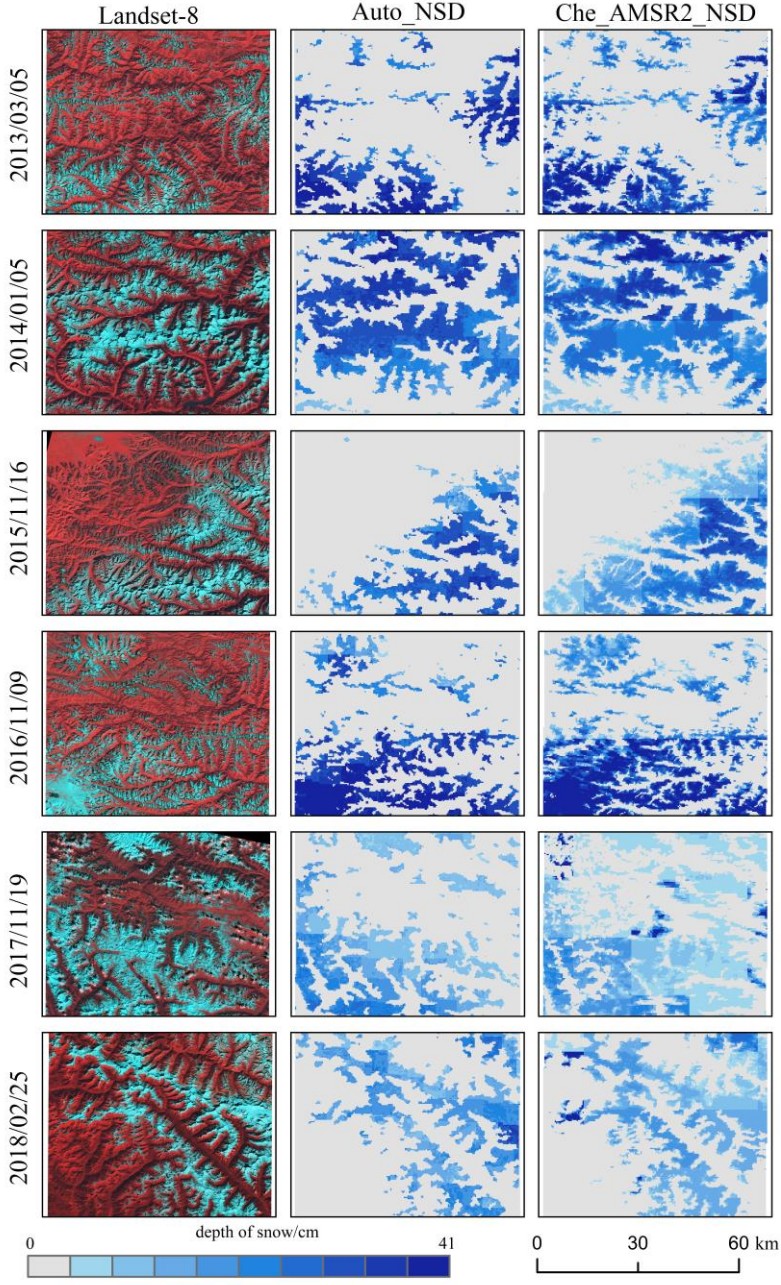


**Figure 10: Comparison of the spatial distribution of Landsat-8 false-color composite images, SD estimation based on AutoML, and downscale SD estimation in 2012-2018.**





## 4.3 Inverse correlation between temperature and SD

Statistical analysis of the monthly average SD time-series trends was conducted on the basis of the Auto_NSD data for the
snow season from 2012 to 2021 on the QTP. In conjunction with the temperature observation data, further analysis of the temporal changes in SD from the Auto_NSD data was performed, as illustrated in Figure 11. The figure demonstrates that, in general, the SD variations during each snow season (from October to March of the following year) are relatively consistent, exhibiting a trend of initially increasing and then decreasing, which is inversely correlated with the changes in ERA5-Land temperature data, showing a trend of initially decreasing and then increasing. The snow season, defined as the period from
October 2013 to March 2014, was selected for analysis. It was observed that as temperatures increased, the average SD exhibited a gradual decrease. In October 2013, the mean SD was 6.16 cm, with a mean monthly temperature of -2.82°C. As time progressed, the temperature underwent a gradual decline, with the November mean temperature dropping to -11.61°C, and the SD of the temperature measurements increased to 8.71 cm. During the period spanning December 2013 to January 2014, the temperature underwent a significant decline, reaching a nadir of -14.49°C and -14.81°C, respectively. This decline
led to the observation of maximum SD values, which exceeded 9 cm. From February to March, as temperatures gradually increased, the SD continued to decrease, with the March SD decreasing to 5.37 cm. The maximum average SD was observed in December 2013, with an average temperature of -14.49°C. In contrast, the minimum average SD was recorded in October 2020, when temperatures rose above freezing, reaching 0.046°C.

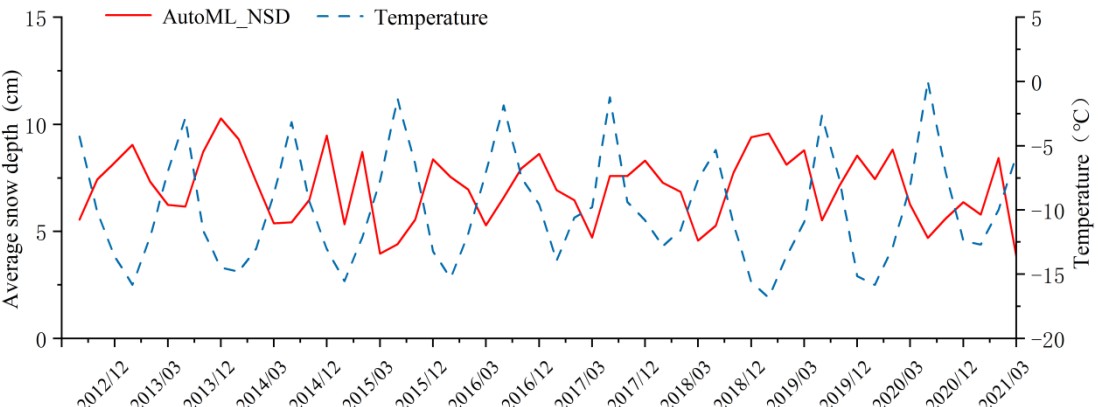

**Figure 11: Trends in Monthly Average SD during the Snow Seasons from 2012 to 2021 on the QTP.**

## 5 Discussion

The QTP is characterised by complex terrain, comprising a mosaic of mountains, plateaus, and basins, resulting in heterogeneous snow distribution and significant interannual variability. It is acknowledged that there are numerous factors influencing SD retrieval, apart from the topographical conditions considered in this study. The accuracy of SD retrieval can
be significantly affected by snow characteristics such as snow density, grain size, liquid water content, as well as vegetation



canopy (Ni et al., 2024; Zhang et al., 2021). In this study, meteorological station data on SD observations were utilised to analyse the SD results obtained from AutoML estimations, considering different SDs, different snow periods and land cover type.

## 5.1 Impact of SD

Research has indicated the possibility of variations in SD retrieval accuracy across different SD ranges (Wang et al., 2022; Wei et al., 2021). The majority of observation stations on the QTP are distributed in shallow snow regions (<10cm). In order to facilitate analysis, this study divides SD into four categories: less than 5 cm, 5-10 cm, 10-20 cm and greater than 20 cm. The accuracy of SD estimation derived from AutoML and downscaled SD methodologies was assessed using meteorological station SD measurements. The results of the study are presented in Table 2. It is evident that when the SD is less than 5 cm,

the Auto_NSD data provides the most accurate SD results, exhibiting the minimum RMSE of 1.69 cm. The bias is 2.71, and the MAE is 2.43 cm, with accuracy evaluation indicators that surpass those of the Che_AMSR2_NSD results. When the SD ranges from 5 to cm, the RMSE of the two SD datasets is 2.94 cm and 6.39 cm, respectively. When the SD exceeds 20 cm, the SD result error reaches its maximum. The RMSE of the Auto_NSD data is 6.43 cm higher than when the SD is less than 5 cm. However, the data comparison results indicate that, irrespective of the SD, the accuracy of SD results based on

AutoML estimation is superior to the Che_AMSR2_NSD results.

It is evident that the accuracy of both SD results is subject to a decline as the standard deviation increases. This phenomenon can be attributed to the continuous increase in SD. Snowmelt may occur on the surface and within the snow layer, leading to alterations in temperature, humidity, and dielectric constant beneath the snow surface. This, in turn, reduces the sensitivity of microwave signals to snow characteristics and affects microwave radiation transmission. The result is an increasing disparity

between the estimated SD and ground SD observation data (Picard et al., 2022; Tanniru and Ramsankaran, 2023; Vuyovich et al., 2017). Concurrently, microwave signals may undergo multiple reflections and scattering within the snow layer, rendering the received signals more complex and unstable(de Gélis et al., 2025). Furthermore, a number of studies have indicated that in instances where the SD exceeds a specified range, saturation issues may emerge in the BT difference between 18 GHz and 36 GHz (10 GHz) (de Gélis et al., 2025; Derksen et al., 2005; Huang et al., 2018).

**Table 2: Accuracy indexes of SD estimation results at different measured SDs.**

| in-situ measurements(cm) | AutoML_NSD | | | Che_AMSR2_NSD | | |
|---|---|---|---|---|---|---|
| | RMSE | BIAS | MAE | RMSE | BIAS | MAE |
| <5 | 1.69 | 2.71 | 2.43 | 4.93 | 2.89 | 3.95 |
| 5-10 | 2.94 | -0.21 | 3.74 | 6.39 | 0.61 | 5.21 |
| 10-20 | 5.13 | 1.47 | 3.80 | 8.36 | -1.07 | 7.29 |
| >20 | 8.12 | 2.31 | 5.89 | 12.13 | -3.26 | 8.26 |



## 5.2 Impact of Snow Accumulation Periods

As time progresses, the properties of snow undergo changes, primarily manifested in variations in snow density and particle size. It is well established that freshly fallen snow is characterised by reduced density and diminished particle size. However, as snow is subjected to changes in temperature and time, it undergoes processes of compaction and ice formation, resulting
in increased density and augmented particle size (Yang et al., 2020b). The present study therefore analysed the influence of differing periods of snow accumulation on the results of SD. The ground-based SD observation data were utilised as the "true value" of SD. The RMSE, BIAS, and MAE between the Auto_NSD and Che_AMSR2_NSD SD results and the "true value" were calculated, as demonstrated in Figure 12. The figure presents a comparative analysis of SD results. It can be observed that, for the corresponding period, the Auto_NSD data demonstrates superior accuracy compared to downscaled
SD data. During the accumulation period, the RMSE of the Auto_NSD model was 4.86 cm lower than that of the Che_AMSR2_NSD model. During the stable snow period, the SD results demonstrate the highest level of accuracy, with RMSE values of 2.11 cm and 5.09 cm, respectively. However, during the snowmelt period, the accuracy of SD results is significantly diminished. The RMSE of the most accurate Auto_NSD data is a mere 5.21 cm, while the less accurate Che_AMSR2_NSD data has an RMSE as high as 12.3 cm.

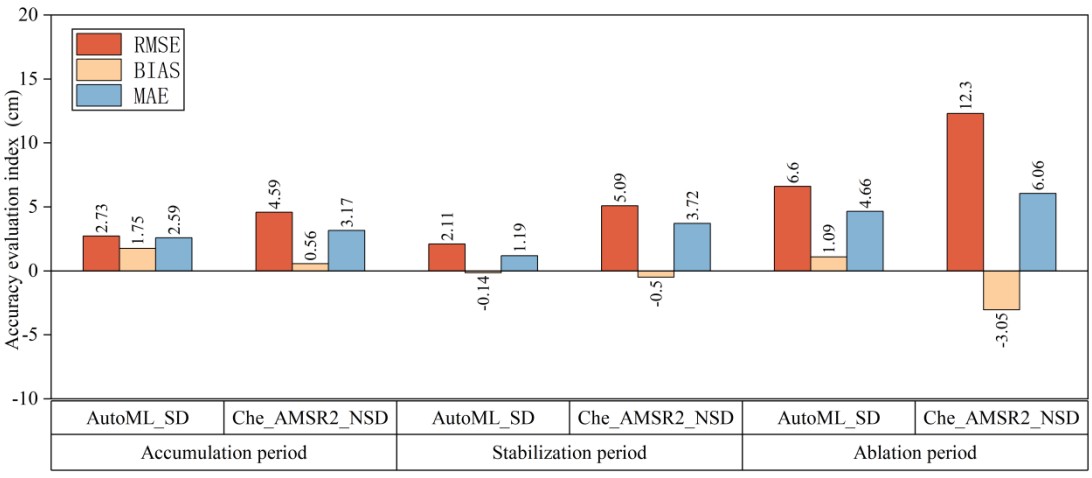


**Figure 12: Comparison of the accuracy of SD data under different snow cover periods with the SD observation data of meteorological stations.**

This discrepancy may be attributed to the fact that during the stable snow period, temperatures are relatively low, snow properties are relatively stable, and changes are minimal, thus exerting less influence on the radiation transmission process
within the snow layer. Conversely, during the snowmelt period, with rising temperatures, snow transitions from dry snow to wet snow, resulting in higher snow density and dielectric constant compared to dry snow, leading to microwave radiation attenuation and weakening of the received signal intensity (Picard et al., 2022; Vuyovich et al., 2017). Moreover, the presence of liquid water in the snow has been shown to enhance the reflection and scattering of microwave signals. This





phenomenon has the potential to impede the penetration of microwave signals to the base of the snow layer, thereby
affecting the accuracy of SD estimation (de Gélis et al., 2025; Yang et al., 2020b).

## 5.3 Impact of Snow Accumulation Periods

The effects of differing land cover types on the accuracy of SD estimation were found to vary. The model simulation results
indicated that the accuracy of SD estimation was lower in forested regions. This is likely due to the attenuation of microwave
radiation received by microwave radiometers from the snow-covered surface layer after it has traversed the vegetation
canopy. This is further compounded by the attenuation of microwave signals to snow caused by the radiation emitted by the
vegetation canopy itself and reflected by the snow layer (Foster et al., 1997; Tanniru and Ramsankaran, 2023).

Whilst the majority of models exhibited satisfactory performance in subaquatic environments, a certain degree of uncertainty
regarding SD estimation persisted. The presence of ice layers in water, which often freeze during the winter months, has
been shown to significantly affect the accuracy of SD estimation. This is due to changes in the physical properties of the
surface, which alter the propagation and reflection of microwave radiation signals. The high dielectric constant and low
reflectivity of ice layers differ significantly from those of snow layers (Cheng et al., 2008). This results in changes to the
path and intensity of the signals, which consequently affects the accuracy of the SD estimation (Newman et al., 2014; Quéno
et al., 2020). Furthermore, the scarcity of ground SD observations in proximity to water bodies within the study may have
led to inaccurate predictions in these regions.

## 5.4 Shortcomings and prospects

The present study has developed a novel methodology for estimating SD in the QTP region. This methodology has been
refined through the use of AutoML techniques. By increasing the spatial resolution of SD data and considering factors
influencing SD inversion, the study established optimal models for four main land cover types, thereby more accurately
representing the heterogeneity of SD pixel distribution. This method has been demonstrated to be particularly efficacious in
the context of rapid SD monitoring in mountainous regions.

Nevertheless, it should be noted that the study is not without its limitations. The distribution of meteorological stations in the
QTP region is characterised by sparsity and heterogeneity. Line-based SD measurements are limited in scope, with a notable
concentration observed in the eastern region. In order to address the non-uniform distribution of ground SD observations, the
approach involved the introduction of downscaled SD data, which was utilised in conjunction with ground SD observations
as training data for AutoML models. Despite the fact that this approach enhanced the precision of SD estimation in the QTP
region to a certain degree, the downscaled SD data itself is inherently uncertain. Consequently, this has led to inconsistencies
between the downscaled and observed SDs. Furthermore, the QTP region displays considerable spatiotemporal heterogeneity
in SD. It is evident that variations in SD distribution are exhibited by different terrains, altitudes and underlying surfaces
during the various stages of the snow season. Despite the study utilising ground SD observations from multiple locations to



verify the accuracy of ML estimates, limitations in observational data coverage precluded the inclusion of further SD observations from diverse terrain regions.

## 6 Conclusion

The present study concentrated on the QTP, employing an ML model with downscaled SD data, ground SD observations, and 19 SD influencing factors as input data. The input data samples were subjected to training under four distinct types of

snow cover (forest, grassland, water, and unused land), and the optimal ML model was selected for each type of snow cover using ten-fold cross-validation. Consequently, the SD sequence data for the QTP from 2012 to 2021 were obtained. A thorough investigation was undertaken to evaluate the precision of Auto_NSD and Che_AMSR2_NSD SD data. This investigation encompassed both quantitative and qualitative analyses, with a focus on a comparison with downscaled SD data. The findings suggested that the SD estimates derived from ML techniques exhibited superior accuracy in characterising

the snow distribution in the QTP region, closely resembling the ground observations, with an R value of 0.81, RMSE of 3.65 cm, BIAS of 0.26 cm, and MAE of 2.62 cm. A comparison with snow cover ranges identified through Landsat-8 imagery demonstrated that both types of SD data were capable of reflecting the detailed spatial features of snow distribution in mountainous regions. However, the Auto_NSD data provided a more consistent description of SD distribution compared to the real SD distribution, fulfilling the monitoring requirements for SD in mountainous regions.


**Code/Data availability:** All code and data are available from the corresponding author upon request.

**Competing interests:** The contact author has declared that none of the authors has any competing interests.

**Author Contributions:** Data curation, Chen Zhang; Funding acquisition, Yanli Zhang; Methodology, Fan Xu; Software, Fan Xu; Supervision, Yanli Zhang; Validation, Xuan Li; Writing – original draft, Fan Xu; Writing – review & editing, Xuan

Li. All authors have read and agreed to the published version of the manuscript.

**Funding:** This research was funded by the National Natural Science Foundation of China (NSFC) project, grant number 42361058.

**Acknowledgments:** We would like to thank the teachers of the Snow Cover Characteristics and Distribution Survey in China for providing the Snow course dataset in typical snow area in China (2017-2019) and SD dataset from common statin

in the typical regions of China during 2017-2019, and the AMSR-2 TB data provided by the Japan Space Agency.

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

        Temperature from Combined MODIS and AMSR2 Data over the Tibetan Plateau. Remote. Sens. 13, 4574.