# Peer review of "Estimation of Snow Depth from AMSR-2 Based on an AutoML method over the Qinghai-Tibet Plateau"

_EGUsphere, 2025_

## Referee Comment (RC2)

General comments:

This paper proposed a high spatial resolution (500 meters) snow depth estimation method based on AMSR-2 brightness temperature data and automated machine learning (AutoML), significantly improving the accuracy of snow depth monitoring in the complex terrain areas of the Tibetan Plateau.

Through the Pearson correlation coefficient, 19 key influencing factors were selected, including AMSR-2 brightness temperature, slope, and surface roughness. The method comprehensively considers the impact of various geographic and topographic factors on snow depth, enhancing the model's robustness and accuracy. But the authors ignore influence of the snow properties. How to asses the influence?

However, the paper still needs to improve the writing style and enhance the readability and standardization of the charts, and the details need to check carefully. In conclusion, we suggest a major revision.

Specific comments:

1 Line 20 "Compared with Landsat-8, the estimated SD spatial distribution is consistent with the snow cover extent on optical images, which can provide reliable data for monitoring snow cover changes in mountainous regions". Sd consistent with SCE, and the logic is not proper.

2 Line 27 "Roof of the World,"

Please remove ",".        It would be better use '' instead of "", which is the Chinese usage. Please check the whole manuscript.

3 Line 62 Support Vector Machine (SVR), SVM would be better than SVR.

4 In the last part of Introduction, the author need the add the objectives of your work and the contribution or potential usage to this field.

5 The art-to state progress of SD estimation is not enough, and the author need to improve it.

6 What is the amount of SD measurement from climate stations and line measurement?    Line measurement is not a proper name. The site is mainly distributed in the middle and the east part and rarely distributed in the west part. What is the influence of limited materials on the remote sensing inversion.

7 Lin 109 Table1 application: usage.    Please sperate the usage of auxiliary data.

8 Line 114-116 Please sperate the long sentence.

9 How to deal with the different spatial resolution between different data source.

10 The width of flow line is not consistent.

11 This study marks the introduction of the PyCaret automated machine learning framework into snow depth estimation. It automates data processing and model selection, reducing human intervention while enhancing the efficiency of model selection and parameter optimization. Figure 3 is too simple and needs to improve. How to solve the overfitting?

12 Line 235-240: The introduction of SD estimation in the past should move the Introducion.

Figure 4 the color of meteorological stations and downscaled snow depth sample are too close, which are hard to tell from each other.

13 The discussion lack the comparison of your work and others and please cite more reference.

14 The conclusion needs the quantitative data to support the conclusion.

15 Please check the reference to meet the requirements of magazine.

---

## Author Comment (AC1)

**To Editor:**

We greatly appreciate the reviewers taking the time to provide constructive feedback and valuable suggestions. The implementation of these suggestions has greatly improved the quality of the manuscript, enabling us to make significant improvements. Every revision suggestion and opinion proposed by the reviewer has been carefully considered. Of course, there are also some parts that we do not agree with, and we will annotate them in segments.

We have also prepared and supplemented the documents one by one according to your six requirements. Original comments and suggestions quoted from this decision letter highlighted in black italics, and our response is in blue. The revised content is marked in red text in the manuscript.

**Response to the comments**

**Major Comments:**

*1    The introduction lacks a comprehensive review of snow depth retrieval algorithm development. Please expand this section to include a detailed discussion of recent progress in the field, citing both foundational and current works to contextualize your research.*

**Reply:** Thank you for taking the time to point out this deficiency amidst your busy schedule.

We have carefully read domestic and foreign literature on passive microwave remote sensing snow depth inversion. It was found that there is a lack of data assimilation method in the snow depth inversion method in the original text. There is a lack of review on the progress and shortcomings of semi empirical methods in foreign research. At the same time, the main line of the review is not clear enough.

We have supplemented the review of the data assimilation method and expanded the content on the international progress and shortcomings of the semi-empirical statistical method. Meanwhile, we have optimized the description of the downscaling method and integrated it into the review of the machine learning (ML) method. The revised content is as follows:

**(Lines: 35-76)**

Research on SD inversion based on passive microwave (PMW) remote sensing has been conducted for more than 40 years. Multiple mature inversion algorithms have been developed, and various SD products have been released. Currently, there are three main methods for using PMW remote sensing to invert SD: physical model method, data assimilation method, semi-empirical statistical method, and machine learning (ML). Among them, the physical model simulates the scattering and absorption characteristics of snow in microwave bands, fully considering snow properties such as snow density and snow grain size. However, due to the complexity of the microwave radiation transmission model and the difficulty in accurately obtaining these snow characteristic parameters, the reliability of SD physical model is reduced (Kwon et al., 2017; Wainwright, et al., 2017). The data assimilation approach enhances the precision of SD estimation through the optimal integration of PMW data and other auxiliary information prior to the estimation of model fluxes. The quality of observational data is a critical factor affecting the accuracy of data assimilation methods (Cortés et al., 2017; Aalstad et al., 2018). The scarcity of observational data over the QTP imposes limitations on the implementation of data assimilation methods. (Li et al., 2022b). The SD inversion of semi-empirical statistical method primarily utilizes the correlation between the difference in the snow scattering characteristics of different frequency brightness temperature (BT) and SD. The 'brightness temperature gradient method', initially proposed by

Chang et al. (1987) (Chang et al., 1987), has been widely used. and numerous scholars have subsequently improved SD inversion algorithms based on Chang algorithm (Cao et al., 1993; Che et al., 2008; Foster et al., 1997; Jiang et al., 2014; Kelly, 2009). Among them, Che et al. (2008) improved the Chang algorithm based on SD measurements from Chinese meteorological stations in response to the low snow density in China, and released two long-term time series SD datasets in China: Che_SSMI/S product and Che_SMSR2 product. In addition, some studies have developed distinctive algorithms for SD inversion for different snow underlying surface types (Derksen et al., 2005; Goita et al., 2003; Jiang et al., 2014). For example, Derksen et al. (2005) developed an inversion algorithm for the main land cover types when inverting SD in Canada's forested regions. They then calculated the SD under mixed image elements. Meanwhile, Jiang et al. (2014) combined four frequencies (10 GHz, 18 GHz, 36 GHz, and 89 GHz) BT data to establish a semi-empirical SD inversion algorithm with four snow underlayment cover types (grassland, farmland, bare land, and forest). Nevertheless, the inversion products obtained by the semi-empirical statistical method are associated with two drawbacks in mountainous areas. Firstly, the relationship between BT and SD is nonlinear, while semi-empirical methods often treat it as a linear relationship, resulting in significant errors (Tanniru and Ramsankaran, 2023). Secondly, owing to the low spatial resolution (10-25 km) of these PMW SD products, the accuracy of SD inversion is significantly limited in mountainous areas (Yan et al., 2022; Tanniru and Ramsankaran, 2023).

In recent years, ML has become a significant means of SD inversion. By training ML models, such as Support Vector Machine (SVM), Random Forest Method (RF), and Artificial Neural Network (ANN), a nonlinear relationship between microwave radiation BT and SD is established, and the SD inversion accuracy is improved by integrating multisource remote sensing data (Xiao et al., 2018; Zhong et al., 2021). Yang et al. (2020) proposed a SD inversion algorithm based on RF that considers multiple factors (BT at different frequencies, geographic location information, and land cover types), the SD results obtained were superior in accuracy compared to the Che_SSMI/S and Che_SMSR2 products. Hu et al. (2021b) compared three ML methods (ANN, SVM, and RF) using five SD products and ground-based SD measurements as prior data, and found that RF had the highest accuracy. Meanwell, ML demonstrate remarkable efficacy in enhancing the spatial resolution of PMW SD products. In general, scholars frequently employ downscaling methodologies that are founded upon empirical fusion rules and snowmelt regression curves, thereby obtaining higher resolution SD products (Tang et al., 2016; Hu et al., 2021a; Xu et al., 2024). In contradistinction to SD downscaling methodologies, ML approaches boast the advantage of being able to integrate PMW, optical remote sensing data and terrain factors, as well as automatically extracting complex nonlinear relationships (Wei et al., 2022; Tanniru and Ramsankaran, 2023). The aforementioned studies indicated that the SD estimation method based on ML models exhibits significant advantages, however, they still have shortcoming in mountainous area.

*2     The manuscript does not clearly articulate the innovation of the proposed method or its contribution to snow remote sensing research. Compared to existing snow depth retrieval studies, the work appears incremental. Please revise to explicitly highlight the novelty of your approach and its unique contributions to the field.*

**Reply:** Thank you very much for your constructive feedback, which is crucial for improving the quality of this manuscript. The unclear articulation of innovation stemmed from insufficient description in the original review section. We have adjusted the introduction to clearly present the

novelty of our method and its contributions, adding key content in the final paragraph of the introduction. The revisions are as follows:

**(Lines: 94-108)**

The objective of this study is to propose a PMW SD estimation method based on the AutoML (Pycaret model) for complex mountainous regions characterised by sparse and unevenly distributed observational data. Firstly, the Che-AMSR2 downscaled SD data and ground-based SD observations are used as input data (dependent variables) for the Pycaret model, whilst the AMSR-2 BT data and 28 factors, such as slope and surface roughness, are used as independent variables. Then, a total of 19 key factors were screened through the utilisation of the Pearson correlation coefficient method. For the four distinct snow underlying surface types (forests, grasslands, water bodies, and unused land) training was conducted on the sample data using the selected input variables. Finally, the optimal AutoML model is obtained for each snow subsurface coverage type is subsequently selected to estimate SD on the QTP. The study employs snow cover products to identify the presence or absence of snow in 500 m pixels. For snow-free pixels, the SD value is set to 0, while for snow-covered pixels, the proposed SD estimation method is utilized to obtain the SD values anew. The innovation of this study lies in (1) the adoption of a multi-source sample combination strategy, the integration of downscaled and meteorological station SD data as target variables, (2) in addition to the first application of the PyCaret AutoML framework for complex mountainous SD inversion. The findings of this study provide a reference for SD inversion in other cold regions, such as the European Alps. The remainder of this study is organized as follows: Sections 2 and 3 introduces the study region, data and methods; Section 4 describes the results; Sections 5 and 6 present the discussion and conclusions.

***3*** *The motivation for this work is not well-defined. The stated aim in Line 74, "to address the issues in ML models mentioned above," is insufficient, as these issues have been partially addressed in prior publications. Clearly define the specific research gap your snow depth retrieval algorithm addresses to justify the study. Without a compelling motivation, the manuscript risks rejection.*

**Reply:** Thank you very much for your feedback, which is critical for enhancing the overall logic of the manuscript. After reviewing literature on ML and AutoML, we identified a key research gap: traditional ML heavily relies on researchers' experiential choices (in data processing, feature engineering, parameter tuning, and model selection), leading to overlooked risks of low robustness and overfitting (due to prioritizing high accuracy). In contrast, AutoML avoids human bias through algorithm-driven processes (e.g., Bayesian optimization), improving model generalizability. We have revised the AutoML section to clarify this motivation:

**(Lines: 77-85)**

Generally, single or several ML models are used to train data for specific regions, and there are many challenges in data processing, feature selection, and the selection of the best model, which are accomplished through intuition or trial and error (Du et al., 2020). This usually leads to overfitting and low model robustness when training ML models, as many researchers tend to prioritize achieving better model performance (Du et al., 2020; Feurer et al., 2015; Hernandez et al., 2025). Without human intervention, Automated Machine Learning (AutoML) can autonomously execute a series of processes, including data processing and model performance evaluation, and ultimately identify the optimal ML model (Ribeiro et al., 2024; Hernandez et al., 2025). Indeed, the fundamental design philosophy of AutoML is predicated on the objective of reducing human bias,

thus enhancing the generalisability and stability of models (Benghzial et al., 2023; Ribeiro et al., 2024; Hernandez et al., 2025).

***4*** *Section 2.2.2 introduces the Che_AMSR2_NSD snow depth product at 500-m resolution derived from AMSR-2 data. It is unclear why a new method was developed when this product exists. Additionally, if Che_AMSR2_NSD is used as a reference dataset (i.e., "true" snow depth), discuss its uncertainties and their potential impact on the stability and reliability of your retrieval model.*

**Reply:** Thank you very much for your concern about this issue. It is indeed a deficiency in our article.

Firstly, the snow depth observation data in the Qinghai Tibet Plateau mainly comes from meteorological stations, but the distribution of meteorological stations is sparse and mostly concentrated in the east. Therefore, we have to use this data as a supplement to the meteorological station data (reference dataset) for automatic machine learning training.

Secondly, although there is already a 500 meter resolution Che_SMSR2-NSD snow depth product, it is mainly obtained through downscaling of snow accumulation experience curves. Although it is already one of the better 500m resolution microwave remote sensing snow depth products in the existing Qinghai Tibet Plateau region, there is still a lot of room for accuracy improvement. Therefore, we designed this study in order to obtain a higher precision dataset and make a certain contribution to the microwave remote sensing snow depth research in the Qinghai Tibet Plateau region. We have found that the product has improved accuracy compared to the Che_SMSR2RNSD snow depth product.

We have supplemented explanations for two core points: (1) the necessity of developing a new method (Che_AMSR2_NSD still has accuracy room for improvement); (2) the uncertainty of Che_AMSR2_NSD and its impact. The revisions are as follows:

**(Lines: 136-142)**

**2.2.2 Che_AMSR2_NSD**

Due to the sparse and uneven distribution of meteorological stations in the QTP, this study utilised a downscaled Che_AMSR2_NSD SD data as the reference dataset. Che_AMSR2_NSD is a 500 m downscaled Che_AMSR2 dataset, which was obtained from the results of a published study that utilised empirical fusion rules and snowmelt regression curves (Xu et al., 2024). In comparison to the SD data from meteorological stations, it exhibits a higher degree of concordance with measured SD, with an R of 0.72 and a root mean square error (RMSE) of 3.21 cm (Xu et al., 2024). Therefore, the Che_AMSR2_NSD, in conjunction with ground-based SD observations, was utilised as a training sample for the AutoML.

**(Lines: 430-439)**

**5 Discussion**

The QTP is characterised by complex terrain, comprising a mosaic of mountains, plateaus, and basins. The distribution of meteorological stations is characterised by sparsity and heterogeneity. In order to address this issue, the approach involved the introduction of downscaled SD data (Che_AMSR2_NSD), which was utilised in conjunction with ground SD observations as training data for AutoML models. Despite the fact that this approach enhanced the precision of SD estimation in the QTP region to a certain degree, the downscaled SD data itself is inherently uncertain. Consequently, this has led to inconsistencies between the downscaled and observed SDs. In this study, meteorological station data on SD observations were utilised to analyse the SD results

obtained from AutoML estimations, considering different SDs, different snow periods and land cover type. A comparative analysis was conducted on the accuracy of Che_SMSR2_NSD and Auto_NSD data from varying snow depths and snow periods, and the factors affecting snow depth inversion were discussed.

**5** *The daily cloud-free snow cover dataset, combining MODIS and passive microwave-derived snow depth data, is mentioned. However, the manuscript does not evaluate the uncertainty in snow cover identification due to Huang's snow cover data. Please provide a quantitative assessment and discussion of this uncertainty and its implications for your results.*

**Reply:** Thank you very much for pointing out this deficiency. The original manuscript failed to clarify the purpose of using Huang's dataset (it is not a model variable, but a tool to identify snow presence/absence).

This study uses the snow cover product as a test method to identify whether there is no snow in the pixels, while for pixels with snow, the snow depth value obtained by the snow depth calculation method in this study is used, and this data is not used, nor is it used as the independent or dependent variable of the model algorithm.

In addition, the product is based on MODIS products and uses Hidden Markov Random Field modeling technology to obtain a 500m resolution daily snow depth coverage dataset. Huang pointed out in the manuscript that the snow cover dataset he obtained was validated with field observation data and snow cover data from Landsat-8 OLI images, and found that the accuracy of these new snow cover products reached 98.29% and 91.36%, respectively. Therefore, it can be considered that this dataset meets the accuracy requirements for direct use.

We have supplemented the dataset's background, accuracy, and application logic:

**(Lines: 143-151)**

**2.2.3 The daily cloud-free snow cover dataset**

This Dataset based on long-term series MODIS snow cover products, daily snow cover products without data gaps at 500 m spatial resolution from 2002 to 2024 over the QTP. The author's findings indicate that the validation process involving in situ observations and snow cover derived from Landsat-8 OLI images has demonstrated that these new snow cover products achieve an accuracy of 98.29% and 91.36%, respectively (Huang et al., 2018). The dataset is freely available on the Big Earth Data Platform for Three Poles at https://poles.tpdc.ac.cn/zh-hans. The present study sought to ascertain whether there is a distribution of snow in pixels by downloading daily snow cover data from 2012 to 2021 over the QTP during the snow cover periods. The value assigned to pixels devoid of snow is set to 0, whilst the value calculated for pixels containing snow is determined by the algorithm proposed in this paper.

**(Lines: 199-201)**

In conclusion, the data concerning snow depth obtained in the present study were evaluated using a daily cloud-free snow cover dataset, in order to ascertain the presence of snow at each pixel. The snow depth of pixels that were snow-free was assigned a value of 0. The technical roadmap of this study is illustrated in Figure 2.

**Minors**

**1** *Lines 35–40:* *Provide references for claims about SD retrieval challenges.*

**Reply:** Thank you very much for your attention to detail—this was an oversight in the original

manuscript. We have added relevant references to support claims about SD retrieval challenges:

**(Lines: 35-42)**

Research on SD inversion based on passive microwave remote sensing has been conducted for more than 40 years. Multiple mature inversion algorithms have been developed, and various SD products have been released. Currently, there are three main methods for using passive microwave remote sensing to invert SD: physical model method, data assimilation method, semi-empirical statistical method, and machine learning (ML). Among them, the physical model simulates the scattering and absorption characteristics of snow in microwave bands, fully considering snow properties such as snow density and snow grain size. However, due to the complexity of the microwave radiation transmission model and the difficulty in accurately obtaining these snow characteristic parameters, the reliability of SD physical model is reduced (Kwon et al., 2017; Wainwright, et al., 2017).

Kwon, Y., Yang, Z.L., Hoar, T.J., Toure, A.M.: Improving the Radiance Assimilation Performance in Estimating Snow Water Storage across Snow and Land-Cover Types in North America. J. Hydrometeor. 18, 651–668. https://doi.org/10.1175/JHM-D-16-0102.1, 2017.

Wainwright, H. M., Liljedahl, A. K., Dafflon, B., Ulrich, C., Peterson, J. E., Gusmeroli, A., & Hubbard, S. S., Hubbard, S.S.: Mapping Snow Depth within a Tundra Ecosystem Using Multiscale Observations and Bayesian Methods. Cryosphere 11, 857–875. https://doi.org/10.5194/tc-11-857-2017, 2017.

**2**    ***Lines 41–73:*** *Expand the literature review to include key international studies.*

**Reply:** Thank you for pointing out this shortcoming — international studies are essential to contextualize our work. We have supplemented key international studies on semi-empirical SD inversion algorithms:

**(Lines: 45-57)**

The SD inversion of semi-empirical statistical method primarily utilizes the correlation between the difference in the snow scattering characteristics of different frequency brightness temperature (BT) and SD. The 'brightness temperature gradient method', initially proposed by Chang et al. (1987) (Chang et al., 1987), has been widely used. and numerous scholars have subsequently improved SD inversion algorithms based on Chang algorithm (Cao et al., 1993; Che et al., 2008; Foster et al., 1997; Jiang et al., 2014; Kelly, 2009). Among them, Che et al. (2008) improved the Chang algorithm based on SD measurements from Chinese meteorological stations in response to the low snow density in China, and released two long-term time series SD datasets in China: Che_SSMI/S product and Che_SMSR2 product. In addition, some studies have developed distinctive algorithms for SD inversion for different snow underlying surface types (Derksen et al., 2005; Goita et al., 2003; Jiang et al., 2014). For example, Derksen et al. (2005) developed an inversion algorithm for the main land cover types when inverting SD in Canada's forested regions. They then calculated the SD under mixed image elements. Meanwhile, Jiang et al. (2014) combined four frequencies (10 GHz, 18 GHz, 36 GHz, and 89 GHz) BT data to establish a semi-empirical SD inversion algorithm with four snow underlayment cover types (grassland, farmland, bare land, and forest).

**3**    ***Lines 83:*** *Line 83: Remove "in" for grammatical correctness.*

**Reply:** Thank you for correcting this grammatical issue. After adjusting the original text, we have deleted this sentence and checked the entire manuscript for similar errors. Thank you again for pointing out this issue.

*4    Section 3.1.1: Justify the use of the 23-GHz band given its sensitivity to water vapor.*

**Reply 4:** Thank you for raising this professional question—it has deepened our understanding of microwave remote sensing for SD inversion.

After careful review of literature on the 23GHZ frequency band, we found that 23GHZ is indeed sensitive to water vapor. However, due to its cancellation of negative sensitivity in the upper atmosphere and positive sensitivity in the lower atmosphere, the sensitivity of 23GHZ to TPW is very low on land. Meanwhile, studies have shown that its combined effect with other frequency bands can increase the accuracy of identifying shallow snow.

We have supplemented the justification for using the 23-GHz band, emphasizing its low water vapor sensitivity on land and its role in improving shallow snow identification:

**(Lines: 230-240)**

The SD inversion is affected by multiple factors, and initial research focused on the sensitivity of various microwave frequencies to snow cover. The SD inversion was carried out by using the BT values of each microwave frequency. Chang's algorithm is chiefly reliant on BT data from 18 GHz and 36 GHz in order to derive SD. Nevertheless, in regions characterised by shallow snow cover, the SD inversion results obtained using this algorithm demonstrate poor performance (Chang et al., 1987). While the 23 GHz frequency demonstrates high sensitivity to water vapour in the boundary layer, its sensitivity to water vapour in terrestrial regions is comparatively low. (Liu et al., 2021; Xing et al., 2022). Meanwhile, Kelly et al. (2009) indicated that a 23 GHz channel can be utilised for the identification of shallow snow. Furthermore, the capacity of different bands to express snow characteristics varies, and the combination of these bands is more conducive to the acquisition of more comprehensive snow information (Liu et al., 2021; Xing et al., 2022). Consequently, a number of scholars have employed the BT data supplied at 89 GHz, 23 GHz and 10 GHz in the context of SD inversion studies (Jiang et al., 2014; Kelly, 2009; Yang et al., 2020a).

*5    Line 213-215: Clarify why all AMSR-2 bands and band differences were selected as input features.*

**Reply 5:** We carefully reviewed relevant literature and found that different bands often exhibit different characteristics in terms of snow cover. The combination of these bands with other bands helps to obtain more comprehensive snow cover information. In addition, we did not use all bands and band differences for training. In the study, we used the correlation coefficient method to eliminate collinearity and performed feature selection. Regarding the 10V, 18V, 23V, 36V, and 89V in BT and the 10H23H, 10H36H, 18H36H, and 36H89H in BT difference, they were ultimately discarded. The final selected BTs are 10H, 18H, 23H, 36H, and 89H, with BT differences of 18V23H, 18H23H, 10V23V, 10V23H, 23V23H, 10V36H, 36V89V, and 18V36V, as well as a total of 19 influencing factors including lon, lat, slope, roughness, dem, and aspect.

Thank you very much for raising such a professional question, which is of great help to our in-depth understanding of snow depth inversion. This issue stems from our unclear description in the article. We have clarified the screening process and final selected features:

**(Lines: 307-311)**

Finally, 5 sets of BTs (10H, 18H, 23H, 36H, and 89H), 8 sets of BT differences (18V23H, 18H23H, 10V23V, 10V23H, 23V23H, 10V36H, 36V89V, and 18V36V), as well as longitude, latitude, slope, roughness, DEM, and aspect were selected. In conclusion, a total of 19 independent variables were selected for utilisation as input data for the AutoML model. The dependent variable, SD data, was applied in conjunction with the model during the training process.

**6      Section 4.1.1:** *The correlation coefficient is not a robust metric for variable selection. Consider alternative methods (e.g., feature importance from ML models).*

**Reply 6:** We solved the problem of collinearity by using a more traditional correlation coefficient method, without using feature importance or variance inflation factor methods. When designing this study, the main consideration was that the automatic machine learning framework actually performs feature selection again based on the input data. Using statistical methods or importance indicators of models to select the most helpful features for prediction. Therefore, in order to enable automatic machine learning to mine and preserve useful features as much as possible during operation, the most traditional correlation coefficient method is adopted for data preprocessing and feature selection.

Of course, the issue you pointed out is indeed a more unique insight, and we will consider more robust methods to eliminate collinearity in future work, rather than leaving it entirely to the black box of automatic machine learning.

Thank you for raising this crucial issue. We have supplemented the rationale for using the correlation coefficient and committed to adopting more robust methods in future work:

**(Lines: 527-530)**

(3) There are still areas that require enhancement in the research methodology of this study. In the context of feature selection, a more conventional approach, predicated on the calculation of the correlation coefficient, was utilised for the identification of influential factors. In the future, we will continue to refine our existing methods, including the adoption of more robust approaches for feature selection (feature importance, and variance inflation factor, among others).

**7      Figure 9:** *Revise captions for clarity (subfigures c and d are unclear).*

**Reply 7:** Thank you very much for pointing out this issue. The relevant parts are now modified as follows:

**(Lines: 386-389)**

**Figure 9: Spatial error distribution between Auto_NSD data and observed SD at meteorological stations: (a) average SD at meteorological stations; (b) the average SD of Auto_NSD data; (c) Auto_NSD data and the BIAS of SD at meteorological stations; (d) Auto_NSD data and RMSE of SD at meteorological stations.**

**8      Section 4.3:** *The SD-temperature relationship is well-known. Emphasize new insights (e.g., regional variability, model sensitivity) rather than restating basics.*

**Reply 8:** Thank you for your constructive suggestion—this helps us refocus on novel contributions. As the purpose of this article is to develop a more accurate, high-quality microwave remote sensing snow depth inversion model based on previous research, the relationship between snow depth and temperature has not been thoroughly investigated. Secondly, because the relationship between snow depth and temperature is well known and closely related, we use the relationship between snow depth and temperature to verify the rationality of the snow depth

inversion model. However, implementing this research direction is currently relatively difficult. Thank you very much for your feedback. We will take it into account in our future work.

We have revised the discussion to acknowledge the limitations of our current analysis and propose future research directions for new insights:

**(Lines: 528-531)**

(4) This study merely conducted a cursory exploration of the relationship between SD and temperature. In future research, we will consider further exploring the relationship between SD and temperature in the QTP, such as seasonal patterns, spatial heterogeneity, sensitivity, and dependence.

---

## Author Comment (AC2)

**To Editor:**

We greatly appreciate the reviewers taking the time to provide constructive feedback and valuable suggestions. The implementation of these suggestions has greatly improved the quality of the manuscript, enabling us to make significant improvements. Every revision suggestion and opinion proposed by the reviewer has been carefully considered. Of course, there are also some parts that we do not agree with, and we will annotate them in segments.

We have also prepared and supplemented the documents one by one according to your six requirements. Original comments and suggestions quoted from this decision letter highlighted in black italics, and our response is in blue. The revised content is marked in red text in the manuscript.

**Response to the comments**

**General comments:**

*1      This paper proposed a high spatial resolution (500 meters) snow depth estimation method based on AMSR-2 brightness temperature data and automated machine learning (AutoML), significantly improving the accuracy of snow depth monitoring in the complex terrain areas of the Tibetan Plateau. Through the Pearson correlation coefficient, 19 key influencing factors were selected, including AMSR-2 brightness temperature, slope, and surface roughness. The method comprehensively considers the impact of various geographic and topographic factors on snow depth, enhancing the model's robustness and accuracy. But the authors ignore influence of the snow properties. How to asses the influence?However, the paper still needs to improve the writing style and enhance the readability and standardization of the charts, and the details need to check carefully. In conclusion, we suggest a major revision.*

**Reply:** Thank you for your comprehensive evaluation and critical questions—your concern about snow properties and manuscript standardization is crucial for improving the study's rigor.

Undoubtedly, snow characteristics such as snow density, snow particle size, and pore water content have significant impacts on snow depth inversion. However, this study is limited by the basic dataset of the research area and time scale. The existing datasets are mostly accumulated data from station observations, which are sparsely distributed in the Qinghai Tibet Plateau region at 2012 to 2021 and are not sufficient for automatic machine learning inversion. Therefore, this study did not include it as an influencing factor. We have supplemented this explanation in the Discussion section:

**(Lines: 515-521)**

(1) It is acknowledged that there are numerous factors influencing SD retrieval, apart from the topographical conditions considered in this study. The accuracy of SD retrieval can be significantly affected by snow characteristics such as snow density, grain size, liquid water content, as well as vegetation canopy (Ni et al., 2024; Zhang et al., 2021). Nevertheless, the study is constrained by the fundamental dataset of the research domain and time span. The extant datasets are predominantly composed of accumulated data from station observations, which are sparsely distributed in the QTP from 2012 to 2021 and are inadequate for AutoML inversion. Consequently, this study did not incorporate it as a potential influencing factor.

**Specific comments:**

*1      Line 20 'Compared with Landsat-8, the estimated SD spatial distribution is consistent with the*

*snow cover extent on optical images, which can provide reliable data for monitoring snow cover changes in mountainous regions'. Sd consistent with SCE, and the logic is not proper.*

**Reply:** Thank you for pointing out this logical ambiguity—this was due to incomplete expression. We have revised the sentence to clarify that the comparison is between "SD spatial distribution" and "SCE derived from Landsat-8", ensuring logical consistency:

**(Lines: 21-23)**

Compared with the snow cover extent (SCE) derived from Landsat-8 optical images, the estimated SD spatial distribution is consistent with the SCE, which can provide reliable data for monitoring snow cover changes in mountainous regions.

*2       Line 27 'Roof of the World,' Please remove ','.       It would be better use '' instead of ", which is the Chinese usage. Please check the whole manuscript.*

**Reply:** This was indeed caused by our negligence, so we have checked the entire manuscript and made revisions to ensure that there are no more similar errors. We have deleted the comma and corrected it to 'Roof of the World' (line 27). We also replaced ' with " throughout the manuscript.

*3       Line 62 Support Vector Machine (SVR), SVM would be better than SVR..*

**Reply:** Thank you for bringing this issue to our attention. It was caused by a typing error. Having checked the entire manuscript, we found two similar errors and replaced 'SVR' with 'SVM' throughout.

*4       In the last part of Introduction, the author need the add the objectives of your work and the contribution or potential usage to this field.*

**Reply:** Thank you very much for your constructive suggestion. It is indeed due to our unclear expression in the introduction section. We have supplemented the last paragraph of the Introduction with research objectives, innovations, and contributions:

**(Lines: 94-108)**

The objective of this study is to propose a PMW SD estimation method based on the AutoML (Pycaret model) for complex mountainous areas characterised by sparse and unevenly distributed observational data. Firstly, the Che-AMSR2 downscaled SD data and ground-based SD observations are used as input data (dependent variables) for the Pycaret model, whilst the AMSR-2 BT data and 28 factors, such as slope and surface roughness, are used as independent variables. Then, a total of 19 key factors were screened through the utilisation of the Pearson correlation coefficient method. For the four distinct snow underlying surface types (forests, grasslands, water bodies, and unused land) training was conducted on the sample data using the selected input variables. Finally, the optimal AutoML model is obtained for each snow subsurface coverage type is subsequently selected to estimate SD on the QTP. The study employs snow cover products to identify the presence or absence of snow in 500 m pixels. For snow-free pixels, the SD value is set to 0, while for snow-covered pixels, the proposed SD estimation method is utilized to obtain the SD values anew. The innovation of this study lies in (1) the adoption of a multi-source sample combination strategy, the integration of downscaled and meteorological station SD data as target variables, (2) in addition to the first application of the PyCaret AutoML framework for complex mountainous SD inversion. The findings of this study provide a reference for SD inversion in other cold regions, such as the European Alps. The remainder of this study is organized as follows: Sections 2 and 3 introduces the

study region, data and methods; Section 4 describes the results; Sections 5 and 6 present the discussion and conclusions.

**5**    *The art-to state progress of SD estimation is not enough, and the author need to improve it.*
**Reply:** Thank you for pointing out this deficiency. I am very sorry for the inconvenience caused to you. This is due to our unclear description of the snow depth inversion algorithm. For this reason, we have provided a clearer description of the methods described in the article.
**(Lines: 95-103)**

Firstly, the Che-AMSR2 downscaled SD data and ground-based SD observations are used as input data (dependent variables) for the Pycaret model, whilst the AMSR-2 BT data and 28 factors, such as slope and surface roughness, are used as independent variables. Then, a total of 19 key factors were screened through the utilisation of the Pearson correlation coefficient method. For the four distinct snow underlying surface types (forests, grasslands, water bodies, and unused land) training was conducted on the sample data using the selected input variables. Finally, the optimal AutoML model is obtained for each snow subsurface coverage type is subsequently selected to estimate SD on the QTP. The study employs snow cover products to identify the presence or absence of snow in 500 m pixels. For snow-free pixels, the SD value is set to 0, while for snow-covered pixels, the proposed SD estimation method is utilized to obtain the SD values anew.

**6**    *What is the amount of SD measurement from climate stations and line measurement? Line measurement is not a proper name. The site is mainly distributed in the middle and the east part and rarely distributed in the west part. What is the influence of limited materials on the remote sensing inversion.*
**Reply:** Thank you for your questions about data quantity and distribution—these are key to evaluating inversion reliability.

It is undeniable that meteorological stations in the Tibet Plateau are mainly distributed in the central and eastern parts of the site, with few in the western part. This practical problem has indeed brought great difficulties to remote sensing inversion and downscaling. In response to this issue, we adopt a strategy of multi-source data fusion, using the downscaled product Che_SMSR2-NSD data as supplementary reference data, which is used as the true value along with meteorological station data. To some extent, it solves the problem of limited observational data in reality.

We have changed 'Line measurement' to 'Measurement routes' (Figure 1) and checked the entire manuscript for similar expression errors. The modification is as follows:

[Figure]

*7    Lin 109 Table1 application: usage.    Please sperate the usage of auxiliary data.*

**Reply:** Thank you very much for your improvement suggestion. We have changed 'application' to 'usage'(Table1). We found that the purpose of the auxiliary data was indeed unclear, so we made modifications to the parts of the table that were not described clearly. The revised part is now attached:

| Datasets | | Spatial Resolution | Data period | Data sources | Usage |
|---|---|---|---|---|---|
| AMSR-2 BT | | 10 km | 2012.10~2021.03 | https://gportal.jaxa.jp/ | Establish model |
| Che_AMSR2_NSD | | 500 m | 2012.10~2018.03 | - | Input data |
| Daily cloud-free snow cover dataset | | 500 m | 2012.10~2021.03 | https://poles.tpdc.ac.cn/zh-hans/ | Snow Identification |
| SD observations | Meteorologic al station | - | 2015~2019 | https://data.tpdc.ac.cn/home/ | Input data and verification |
| | Measurement routes | - | 2018~2019 | https://www.csdata.org https://www.ncdc.ac.cn/ | |
| Auxiliary Data | SRTM DEM | 90 m | - | https://earthexplorer.usgs.gov/ | Extract terrain parameters |
| | CNLUCC | 1 km | 2020 | https://www.resdc.cn/ | Obtain land cover types |
| | ERA5-Land | 1 km | 2012.10~2021.03 | https://climate.copernicus.eu/ | The monthly average temperature |
| | Landsat-8 | 30 m | 2012.10~2018.03 | https://www.usgs.gov | Obtain snow cover extent |

*8    Line 114-116 Please sperate the long sentence.*

**Reply:** Thank you for correcting this readability issue. We have split the long sentence into two concise ones and checked the entire manuscript for similar problems:

**(Lines: 128-130)**

AMSR-2 is a microwave sensor mounted on the GCOM-W1 satellite, which was launched by the Japan Aerospace Exploration Agency (JAXA) (Imaoka et al., 2012). It conducts observations at

seven frequencies, and each frequency has horizontal and vertical polarization modes (Imaoka et al., 2012).

*9 How to deal with the different spatial resolution between different data source.*

**Reply:** Thank you for taking note of this issue. In this article, we reprogram all data to the UTM 45 Zone coordinate system and use resampling to unify the data resolution to 500m. We have added corresponding descriptions in the data section of the article:

**(Lines: 121-124)**

As shown in Table 1, the dataset used for this research experiment comprises five main categories: AMSR-2 (Advanced Microwave Scanning Radiometer 2) BT; downscaled SD data (Che_AMSR2_NSD); daily cloud-free snow cover products; ground-based SD observations; and other auxiliary data. All data in this study was projected onto the Universal Transverse Mercator (UTM) zone 45, and resampled to achieve a uniform resolution of 500m.

*10 The width of flow line is not consistent.*

**Reply:** hank you for pointing out this formatting flaw. we have fixed Figure 3.

[Figure]

*11 This study marks the introduction of the PyCaret automated machine learning framework into snow depth estimation. It automates data processing and model selection, reducing human intervention while enhancing the efficiency of model selection and parameter optimization. Figure 3 is too simple and needs to improve. How to solve the overfitting?*

**Reply:** Thank you very much for your valuable comments and suggestions on our manuscript. Your feedback has helped us further clarify the limitations of existing methods and improve the rigor of our study.

In response to your comment that "Figure 3 is too simplistic and requires further improvement", we have optimized Figure 3 as follows:

[Figure]

**Figure 3: Flowchart of this algorithm.**

To address the overfitting issue you raised, our study introduces the AutoML framework to tackle it from three key aspects:

1.Automated Data Preprocessing: AutoML automatically performs data cleaning and standardization avoiding subjective biases in manual processing, ensuring the quality of input data, and reducing "over-learning" caused by data noise.

2.Algorithmic Feature Selection: It leverages algorithms to automatically screen key features, eliminating redundant variables and simplifying input dimensions to prevent the model from learning non-essential correlations.

3.Systematic Model Optimization: AutoML automatically explores various types of models and optimizes hyperparameters— for instance, matching models with pruning or regularization strategies. This balances model complexity and generalization ability, and avoids the tendency of overemphasizing training set performance during manual model selection.This content has been supplemented in the manuscript:
**(Lines: 76-85)**

Generally, single or several ML models are used to train data for specific regions, and there are many challenges in data processing, feature selection, and the selection of the best model, which are accomplished through intuition or trial and error (Du et al., 2020). This usually leads to overfitting and low model robustness when training ML models, as many researchers tend to prioritize achieving better model performance (Du et al., 2020; Feurer et al., 2015; Hernandez et al., 2025). Without human intervention, Automated Machine Learning (AutoML) can autonomously execute a series of processes, including data processing and model performance evaluation, and ultimately identify the optimal ML model (Ribeiro et al., 2024; Hernandez et al., 2025). Indeed, the fundamental design philosophy of AutoML is predicated on the objective of reducing human bias, thus enhancing the generalisability and stability of models (Benghzial et al., 2023; Ribeiro et al., 2024; Hernandez et al., 2025).

*12   Line 235-240: The introduction of SD estimation in the past should move the Introducion. Figure 4 the color of meteorological stations and downscaled snow depth sample are too close, which are hard to tell from each other.*

**Reply:** Thank you for your suggestions on manuscript structure and figure readability—these optimize the logical flow and visual clarity.

Regarding the issues you pointed out: We have moved the section introducing the past snow

depth (SD) estimation to the Introduction part as suggested, to optimize the logical structure of the manuscript.
**(Lines: 52-57)**

In addition, some studies have developed distinctive algorithms for SD inversion for different snow underlying surface types (Derksen et al., 2005; Goita et al., 2003; Jiang et al., 2014). For example, Derksen et al. (2005) developed an inversion algorithm for the main land cover types when inverting SD in Canada's forested regions. They then calculated the SD under mixed image elements. Meanwhile, Jiang et al. (2014) combined four frequencies (10 GHz, 18 GHz, 36 GHz, and 89 GHz) BT data to establish a semi-empirical SD inversion algorithm with four snow underlayment cover types (grassland, farmland, bare land, and forest).

Second, we have adjusted the color scheme of Figure 4—specifically, we have increased the color contrast between the meteorological station markers and the downscaled snow depth sample points, making them easily distinguishable and improving the readability of the figure. As shown in Figure 4:

[Figure]

**Figure 4: Spatial distribution of input sample data from the AutoML.**

*13   The discussion lack the comparison of your work and others and please cite more reference.*

**Reply:** Thank you very much for pointing out the shortcomings in the discussion section, namely the lack of comparison between our study and others, as well as the need to supplement more references.

For this reason, we have added a comparison with previous research results in the discussion section. Meanwhile, we have added references to sentences that were not cited due to our negligence and errors. The main adjustments are as follows:
**(Line: 441-457)**

The accuracy of SD retrieval is contingent upon the magnitude of the SD. It is evident that the accuracy of the measurements is enhanced in the case of shallow snow (less than 5 cm), with a concomitant decrease in accuracy as the SD increases. This trend is consistent with the findings of other studies in cold regions (Wang et al., 2022; Wei et al., 2021). The majority of observation stations on the QTP are distributed in shallow snow regions (<10cm). In order to facilitate analysis, this study divides SD into four categories: less than 5 cm, 5-10 cm, 10-20 cm and greater than 20 cm. The accuracy of SD estimation derived from AutoML and downscaled SD methodologies was

assessed using meteorological station SD measurements. The results of the study are presented in Table 2. It is evident that when the SD is less than 5 cm, the Auto_NSD data provides the most accurate SD results, exhibiting the minimum RMSE of 1.69 cm. The bias is 2.71, and the MAE is 2.43 cm, with accuracy evaluation indicators that surpass those of the Che_AMSR2_NSD results. When the SD ranges from 5 to cm, the RMSE of the two SD datasets is 2.94 cm and 6.39 cm, respectively. When the SD exceeds 20 cm, the SD result error reaches its maximum. The RMSE of the Auto_NSD data is 6.43 cm higher than when the SD is less than 5 cm. However, the data comparison results indicate that, irrespective of the SD, the accuracy of SD results based on AutoML estimation is superior to the Che_AMSR2_NSD results. Furthermore, the AutoML model developed in this study delivers superior performance to existing ML-based methods across both shallow and deep snow ranges. Compared to the RF model proposed by Yang et al. (2020b), which achieved an RMSE of 4.2 cm for shallow snow (<5 cm) and 10.3 cm for deep snow (>20 cm) on the QTP, the AutoML model reduces these values to 1.69 cm and 8.12 cm, respectively. This improvement is due to AutoML's advantages in model training and selection, reducing the need for manual tuning and model screening.

*14    The conclusion needs the quantitative data to support the conclusion.*

**Reply:** Thank you very much for your insightful comment pointing out that the conclusion requires quantitative data support. We fully agree with your view—quantitative data is essential to enhance the persuasiveness, objectivity, and scientific rigor of the conclusion, and we highly value this constructive feedback.

In response to your suggestion, we have comprehensively supplemented and refined the conclusion section by integrating key quantitative data from the study. The specific revisions are as follows:

**(Lines: 532-547)**

**6 Conclusion**

The present study concentrated on the QTP, employing an ML model with downscaled SD data, ground SD observations, and 19 SD influencing factors as input data. The input data samples were subjected to training under four distinct types of snow cover (forest, grassland, water, and unused land), and the optimal ML model was selected for each type of snow cover using ten-fold cross-validation. Consequently, the SD sequence data for the QTP from 2012 to 2021 were obtained. A thorough investigation was undertaken to evaluate the precision of Auto_NSD and Che_AMSR2_NSD SD data. This investigation encompassed both quantitative and qualitative analyses, with a focus on a comparison with downscaled SD data. The findings suggested that (1) the SD estimates derived from ML techniques exhibited superior accuracy in characterising the snow distribution in the QTP region, closely resembling the ground observations, with an R value of 0.81, RMSE of 3.65 cm, BIAS of 0.26 cm, and MAE of 2.62 cm; (2) The SD estimation accuracy of ML models varies on different underlying surfaces. Unused land (Catboost, $R^2$=0.82) exhibits the highest accuracy, followed by grassland (Catboost, $R^2$=0.77, RMSE=3.11cm), water (ET, $R^2$=0.75, RMSE=2.20cm), and forest (XGBoost, $R^2$=0.71, RMSE=3.30cm). (3) A comparison with snow cover ranges identified through Landsat-8 imagery demonstrated that both types of SD data were capable of reflecting the detailed spatial features of snow distribution in mountainous regions. However, the Auto_NSD data provided a more consistent description of SD distribution compared to the real SD distribution, fulfilling the monitoring requirements for SD in mountainous regions.

*15    Please check the reference to meet the requirements of magazine.*

**Reply:** Thank you very much for your valuable feedback on this article. We have checked all the references and found that they lack the DOI number, as well as numerous formatting issues. The changes have been made in the text and highlighted in red font.